



Atmospheric
Chemistry
and Physics

# Ice nucleation properties of K-feldspar polymorphs and plagioclase feldspars

**André Welti[1,a], Ulrike Lohmann[1], and Zamin A. Kanji[1]**

[1]Institute for Atmospheric and Climate Science, ETH Zurich, Zurich, 8092, Switzerland
[a]now at: Finnish Meteorological Institute, Helsinki, 00101, Finland

**Correspondence:** Zamin A. Kanji (zamin.kanji@env.ethz.ch)

**Abstract.** TS1 TS2 CE1The relation between the mineralogical characteristics of size-selected feldspar particles from 50 to 800 nm and their ability to act as ice-nucleating particles (INPs) in the immersion mode is presented. Five polymorph members of K feldspar (two microclines, orthoclase, adularia and sanidine) and four plagioclase samples (three labradorites and a pericline sample) are tested. Microcline was found to be the most active INP in the immersion mode consistent with previous findings. Samples were selected for their differences in typical feldspar properties such as crystal structure, bulk and trace elemental composition, and ordering of the crystal lattice. The properties mentioned are related to the temperature of feldspar crystallization from the magma during formation. Properties characteristic of low-temperature feldspar formation coincide with an increased ability to nucleate ice. Amongst the samples investigated, ice nucleation is most efficient on the crystallographically ordered, triclinic K-feldspar species microcline, while the intermediate and disordered monoclinic K-feldspar polymorphs orthoclase and sanidine nucleate ice at lower temperatures. The ice nucleation ability of disordered triclinic Na/Ca feldspar is comparable to disordered K feldspar. The conditions of feldspar rock formation also leave a chemical fingerprint with varying abundance of trace elements in the samples. X-ray fluorescence spectroscopy analysis was conducted to determine metal oxide and trace elemental composition of the feldspar samples. The analysis revealed a correlation of trace metal abundance with median freezing temperatures ($T_{50}$) of the K-feldspar samples allowing us to sort them for their ice nucleation efficiency according to the abundance of specific trace elements. A pronounced size dependence of ice nucleation activity for the feldspar samples is observed, with the activity of smaller-sized particles scaling with surface area or being even higher compared to larger particles. The size dependence varies for different feldspar samples. In particular, microcline exhibited immersion freezing even for 50 nm particles which is unique for heterogeneous ice nucleation of mineral dusts. This suggests that small microcline particles that are susceptible to long-range transport can affect cloud properties via immersion freezing far away from the source. The measurements generally imply that temperatures at which feldspars can affect cloud glaciation depend on the transported particle size in addition to the abundance of these particles.

## 1 Introduction

Freezing of water droplets in supercooled clouds begins in a number of ways. Cloud droplets with a typical mean diameter of 10 μm freeze by homogeneous nucleation at temperatures below 235 K. Above homogeneous nucleation temperatures, drop freezing is initiated through heterogeneous ice nucleation by inclusions in water droplets (immersion freezing), serving as substrates for ice nucleation or by particle collisions with cloud droplets (contact freezing). Dependent on the type of particle, the probability that heterogeneous ice nucleation occurs at a certain temperature differs. Through the analysis of ice crystal residuals it was discovered that often mineral dust particles serve as ice nucleation substrates (e.g. Kumai, 1951). Rosinski (1979) identified feldspar (orthoclase) as one of the mineral residuals. Feldspar is the most abundant mineral type in the outer crust of the earth (Lutgens et al., 2014) and therefore a common compound of desert

dusts and volcanic ejecta (Knippertz and Stuut, 2014). The feldspar minerals can be grouped, according to bulk composition, into potassium-rich feldspar (K feldspar) and Na- or Ca-rich species (plagioclase). While the series of plagioclase contains all fractions of Na and Ca, there is a mixing gap for K feldspar. Only a limited fraction of $K^+$ can be replaced by $Na^+$ or $Ca^{2+}$. K feldspar is subdivided into three polymorphs (microcline, orthoclase, sanidine), based upon the degree of order–disorder of $Al^{3+}$ and $Si^{4+}$ in the tetrahedral units forming their crystal network. The units are made up by one $Al^{3+}$ and three $Si^{4+}$ ions. In a disordered state $Al^{3+}$ can be found in any one of the four tetrahedral sites while in an ordered state $Al^{3+}$ occupies the same site throughout the crystal (Nesse, 2016).

Early investigations on the ice nucleation efficiencies of various mineral dust species, including feldspar (orthoclase and microcline), by, e.g., Pruppacher and Sänger (1955), Mason and Maybank (1958), Isono and Ikebe (1960), grouped orthoclase amongst the less efficient ice-nucleating particles compared to mineral species, e.g. muscovite or kaolinite. In addition, differences in ice nucleation between K-feldspar species are already recognized; for example, Mason and Maybank (1958) reported microcline to have an 8.5 K higher ice-nucleating threshold temperature than orthoclase. More recent studies (Zimmermann et al., 2008; Atkinson et al., 2013; Yakobi-Hancock et al., 2013; Harrison et al., 2016; Kaufmann et al., 2016; Kiselev et al., 2016; Peckhaus et al., 2016; Pedevilla et al., 2016; Whale et al., 2017) confirmed that different members of the feldspar group exhibit variable ice nucleation activities in the immersion and deposition mode. Samples of microcline, in particular, were found to be active ice-nucleating particles (INPs) at higher temperatures than other minerals. Even though feldspar was recognized as an INP in earlier studies, little attention was given to feldspar in the context of atmospheric ice formation. Instead, ice nucleation research focused on clay minerals, probably because feldspar is mainly found in the coarse size fraction in airborne desert dusts (Engelbrecht and Derbyshire, 2010), making it less susceptible to long-range transport. In support of this, using X-ray diffraction analysis, Boose et al. (2016) found the feldspar mass fraction in airborne transported dusts to be significantly lower than in surface sampled dusts. Despite the lower mass fraction of feldspars found in the airborne Saharan dust samples, Boose et al. (2016) concluded that these could be relevant for atmospheric cloud glaciation at warmer temperatures ($T > 250$ K). Explosive volcanic eruptions can introduce large quantities of feldspar particles into the atmosphere with a regional to global effect on cloud glaciation (Mangan et al., 2017). The analysis of ash particles that make up volcanic ash clouds showed that up to 70 % can be feldspar (usually plagioclase feldspars, rarely sanidine) particles (Bayhurst et al., 1994,; Schumann et al., 2011).

Reasons for why microcline feldspars are active INPs have been discussed in recent studies. A survey of 15 feldspars by

Harrison et al. (2016) confirmed the general notion that K-feldspar suspensions show higher freezing temperatures than plagioclase feldspars. In addition to one specific K-feldspar sample being very active with a freezing onset of $\sim -2\,^\circ$C (for 1 wt % suspension) in the immersion mode, Harrison et al. (2016) showed that one sample of an Na feldspar (Amelia albite) was also particularly ice active, exhibiting freezing onset already at $-4\,^\circ$C (for 1 wt % suspension). They also concluded that not all feldspars are equal for their ice nucleation properties even if they are from the same subgroup. Whale et al. (2017) investigated 15 additional alkali feldspars but found no correlation of their ice nucleation activities with their crystal structures or chemical compositions. Instead, they explored specific topographic features on samples that showed exceptionally high ice nucleation temperatures and found that these samples exhibited certain microtextures (perthitic samples) related to phase separation into Na- and K-rich regions, whereas samples active at lower temperatures were non-perthitic (Whale et al., 2017). Based on results of a molecular model, Pedevilla et al. (2016) show that freshly cleaved microcline (001 surface) can adsorb a monolayer of water possessing a non-ice-like structure; however, overlying second and third layers will have ice-like structures, thus promoting ice nucleation. Using an electron microscope, Kiselev et al. (2016) observed that even along the perfectly cleaved (001) surface of K feldspar, ice nucleation occurred on microscopic sites which expose the high-energy (100) surface. This observation was supported by the alignment of growing ice crystals with the (100) surface, regardless of the orientation of the surface (001 or 010) on which ice nucleation occurred. Additionally, ice nucleation predominantly occurred at sites exhibiting steps, crevices or cracks where the high-energy (100) surface is thought to be exposed.

Feldspars contain a variety of trace elements which can, for example, influence the appearance or colouring of the rock. Some trace elements in feldspars are representative of partitioning during the mineralization process when the rocks form from the parent magma. Microtextures found within feldspar strongly influence the subsequent behaviour of feldspars, for example during low-temperature weathering, and are central to the exchange (or retention) of trace elements (Parsons et al., 2015). Partitioning of trace elements can reveal microtextures, which are related to the distribution of trace elements and point defects that are otherwise difficult to detect in feldspars. Both microtextures and point defects have been suggested to be important for immersion ice nucleation on feldspar polymorphs (Whale et al., 2017). Due to the interlinked nature of feldspar characteristics and trace elemental composition, it may be possible to relate the ice nucleation activity to the concentration of such trace elements within a sample.

In the current study, we present the ice nucleation ability of single immersed, size-selected particles of five K-feldspar polymorphs (see Table 1) with both monoclinic and triclinic crystal structures. Measurements with single immersed par-

**Table 1.** Sample specification of feldspars used in this study.

| Sample name | Source region | Feldspar type | Crystal structure |
|---|---|---|---|
| Orthoclase | Italy (Elba) | K | Monoclinic |
| Adularia | Switzerland | K | Monoclinic |
| Sanidine | Germany | K | Monoclinic |
| Microcline | Italy (Elba) | K | Triclinic |
| Microcline (Amazonite) | Namibia | K | Triclinic |
| Labradorite AU | Austria | Na/Ca | Triclinic |
| Pericline (Albite) | Switzerland | Na | Triclinic |
| Labradorite CH2 | Switzerland | Na/Ca | Triclinic |
| Labradorite CH1 | Switzerland | Na/Ca | Triclinic |

ticles allows us to explore the lower range of ice nucleation temperatures ($< 253$ K) within the K-feldspar group, focusing especially on the effect of particle size. Four Na- and/or Ca-feldspar species are included (see Table 1) to use the feldspar group as a natural system to examine the importance of potential ice-nucleating properties (compositional and mineralogical) which vary slightly among its members.

## 2 Sample description

Figure 1 shows a ternary phase diagram of the feldspar samples used in this study. The samples are grouped according to their proportional bulk K, Ca and Na content found by X-ray fluorescence (WD-XRF; Axios, PANalytical). Where known, the samples are named following the mineralogical nomenclature and specific variety; for example, adularia is a variety of orthoclase and amazonite a variety of microcline. Sample origin, name, mineralogical classification (K feldspar or Na/Ca feldspar) and crystal structure are summarized in Table 1. The naming of the plagioclase samples is based on the compositional analysis. For the plagioclase group we have three labradorite samples with different Ca/Na proportions (and the same origin, Switzerland: CH1 and CH2) or the same Ca/Na proportions but a different origin (CH2 and Austria, AU), and one form of albite, called pericline (see Table 1). Five K-feldspar samples are investigated including the polymorphs microcline, orthoclase and sanidine. Order–disorder polymorphism occurs due to the effect of pressure, temperature, cooling rate and variations in bulk composition on the ordering process of atoms in the crystalline lattice during petrogenesis, i.e. the formation of the mineral by crystallization from the melt (Parsons and Boyd, 1971). The lowest ordered polymorph, sanidine, forms at high temperature, orthoclase at intermediate temperature and microcline is the stable polymorph formed at the lowest crystallization temperature. The different polymorphs differ in some physical properties (e.g. melting point) and are found in different rocks: sanidine in volcanic and very high-temperature metamorphic rocks, orthoclase in volcanic and high-temperature metamorphic rocks, and microcline in

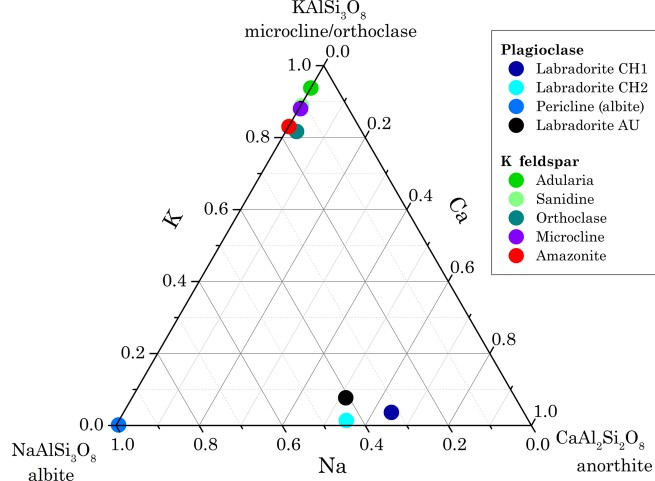

**Figure 1.** Phase diagram of feldspar samples investigated. Composition determined from X-ray fluorescence spectroscopy (XRF).

granitic and metamorphic rocks (Lutgens et al., 2014). Depending on the source region, feldspars vary in secondary chemical and trace elemental composition (see Sect. 5.2). Samples used in this study are not chosen for atmospheric relevance but to investigate the single-particle ice nucleation properties of feldspar particles of different geological origin and interlinked, crystal structure, and bulk and trace elemental composition. As such, some samples studied here like adularia and pericline are rare because they typically come from mineral veins and may not be found as the main constituent of airborne dusts (Lee et al., 1998). Similarly, sanidine would also be less abundant in dusts, except following volcanic ash eruptions. However, orthoclase and microcline samples that originate from granites and gneisses do eventually reach the atmosphere after several cycles of weathering and sedimentation (Lee et al., 1998). Plagioclases such as labradorites are common in basalts and as such can be found in volcanic ash from basaltic eruptions (Lee et al., 1998).

## 3 Experimental method

All feldspar samples were tested for their ice-nucleating ability in the immersion mode by immersing single, size-selected particles of mobility diameters between 50 and 800 nm in water droplets. Figure 2 depicts a schematic of the monodisperse particle generation unit and the ice nucleation experiment. The two stages of the experiment are described below.

### 3.1 Sample preparation

Basic raw materials are single feldspar crystals (adularia, amazonite), one ground sample for use in pottery (labradorite AU) and rocks (all the others). All raw samples are ground for 5 min in a tungsten carbide disc mill (Retsch, RS1, 1400 rpm). The ground sample is aerosolized in a flu-

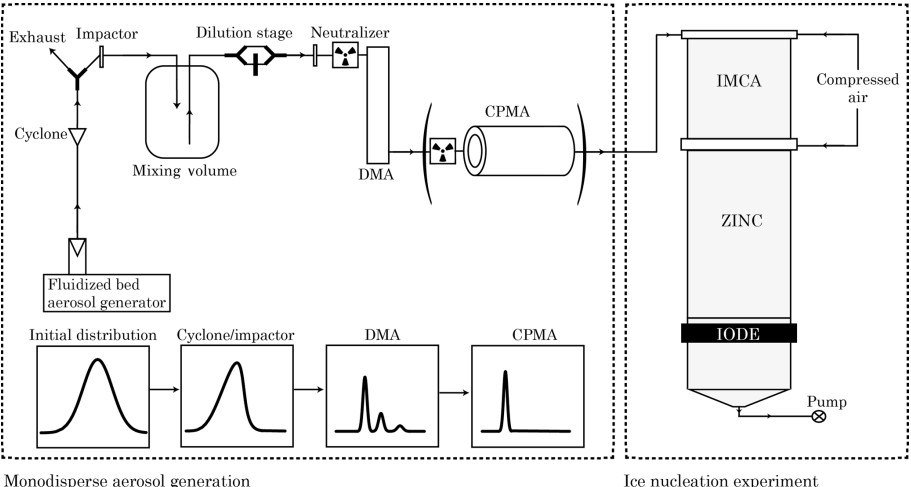

**Figure 2.** Schematic of the particle generation and ice nucleation set-up used to examine immersion freezing of the feldspar samples.

idized bed aerosol generator (TSI Model 3400A). To investigate particle size effects special attention is given to select monodisperse particles. The aerosol is passed through two cyclones ($d_{50} = 3$ and $1\,\mu m$) and one impactor ($d_{50} = 0.4\,\mu m$) before size selection by a differential mobility analyser (DMA; TSI Model 3081). As the size selection with a DMA is not directly based on the particle size but on electrostatic mobility, multiple charged CE2 particles of larger sizes pass the DMA and are selected as well. Depending on the initial particle size distribution, the fraction of multiple charged particles can contribute a substantial fraction to a certain mobility diameter. In particular, small particle sizes originating from the upslope of the initial size distribution contain nonnegligible amounts of multiple charged, larger particles. To reduce distortion to the size selection, a centrifugal particle mass analyser (CPMA; Cambustion), able to isolate a specific particle mass, is used in sequence with the DMA for particle sizes of 100 and 200 nm. Spherical particle shape and bulk density ($2.6\,\mathrm{g\,cm^{-3}}$) was assumed to select singly charged particles and exclude larger particles by mass. The use of the CPMA for the selection of larger particles (400, 800 nm) was not necessary as the fraction of larger particles is sufficiently reduced by the cyclones and the impactor upstream of the DMA (see Fig. 2), which confines the size distribution of the particles entering the DMA to $d_{50} = 1\,\mu m$ (aerodynamic diameter). For the 400 nm size selection, doubly and triply charged particles (700 and 994 nm) amounted to 16 % and 4 %, respectively, whereas for the 800 nm particles, the doubly charged particles (1481 nm) made up 4 %. The CPMA was not used for the selection of 50 nm particles due to the very low number concentration of this particle size in the output size distribution of the fluidized bed generator. A substantial fraction of larger, multiple charged particles (73 nm ($\sim 30$ %) and 91 nm ($\sim 7$ %) for 2 and 3 charges, respectively) among the 50 nm ($\sim 63$ %) particles is expected.

The contributions of singly and multiply charged particles to the frozen fractions is approximately proportional to their abundance and does not impact any of the conclusions made in this work. As all feldspar samples have similar hardness and were ground in the same way, the fraction of multiple charged particles at a selected size is expected to be comparable among samples. The contribution of multiply charged particles to the 50 nm particle frozen fraction is shown in Appendix C.

## 3.2 Immersion freezing experiment

A flow of $1\,\mathrm{L\,min^{-1}}$ of sample air containing the dry feldspar particles layered between twice $2.5\,\mathrm{L\,min^{-1}}$ sheath air is introduced into the immersion mode cooling chamber (IMCA) developed by Lüönd et al. (2010). The IMCA walls are 0.6 cm apart and layered with continuously wetted filter paper. A horizontal and vertical temperature gradient is maintained to generate supersaturated conditions to form and subsequently precondition water droplets for the experiment in the Zurich Ice Nucleation Chamber (ZINC; Stetzer et al., 2008). In the top part of IMCA particles are exposed to relative humidity with respect to water of $\sim 120$ % at 313 K. The high relative humidity ensures all particles transitioning through IMCA activate into cloud droplets, independent of their properties. In the lower part of IMCA, droplets containing the immersed feldspar particles are cooled to the experimental temperatures prevailing in ZINC. ZINC is a parallel-plate continuous-flow diffusion chamber (1 cm gap between the wall plates) layered with a thin ice layer as source of water vapour at temperatures below 273 K. The temperature of the parallel wall plates of the chamber is controlled independently by two cryostats (Lauda134 RP890). At the transition from IMCA into ZINC, additional sheath air ($2\,\mathrm{L\,min^{-1}}$ on either side) is added in order to maintain the 1 mm width of the sample lamina and to prevent turbulence. Constraining

the sample width to 1 mm allows us to control and constrain the temperature and relative humidity that the particles are exposed to. ZINC is operated such that droplets transitioning from IMCA can be exposed to temperatures from 238 to 258 K at water saturation conditions to prevent evaporation of the droplets. After a residence time of ∼ 10 s, water droplets and nucleated ice crystals are detected in-line (see Fig. 2) by the ice optical depolarization detector (IODE) described in Nicolet et al. (2010) and Lüönd et al. (2010). The water droplet and ice crystal counts at a specific temperature are used to derive the fraction of droplets that freeze due to ice nucleation on the immersed feldspar particle or by homogeneous freezing. The reported frozen fraction is defined as

$$\text{frozen fraction} = \frac{\#\,\text{ice crystals}}{\#(\text{ice crystals} + \text{water droplets})}. \quad (1)$$

## 4   Experimental results

### 4.1   Temperature dependence

The shape of the temperature-dependent frozen fraction is important because this dependency can dictate at which cloud supercooling one can expect particles to actively form ice and initiate glaciation in the mixed-phase cloud regime. Comparing frozen fraction shown in Fig. 3 reveals different slopes of the temperature dependence for different samples as well as particle sizes. As an example, the increase in frozen fraction from 0.1 to 1 occurs over a 7 K range for 800 nm microcline particles; however, this range broadens with decreasing particle size to 10 K for 100 nm particles. To illustrate this further, in Fig. 4 we show the range of freezing temperatures for the frozen fraction 0.1–1 for all 800 nm particle samples. We find that the freezing temperature range for orthoclase spreads over 10 K already for the 800 nm particles. The broader range of activation temperatures suggests that the ice-nucleation-active sites in the 800 nm orthoclase are more heterogeneous than the ones in the 800 nm microcline sample (see Fig. 4). Additionally, we observe that the microcline and orthoclase samples initiate freezing at warmer temperatures compared to the Na/Ca feldspars suggesting that K feldspars are in general more ice active in accordance with Atkinson et al. (2013). The microcline amazonite sample is the most effective sample for immersion freezing. Upon close observation of the freezing range depicted in Fig. 4, however, it can be seen that the narrow freezing range is neither unique to microcline nor the K-feldspar polymorphs as a freezing range of 5 K also exhibited by the labradorite CH2 TS3 sample. Even though labradorite CH2 is much less ice active than orthoclase, the narrow spread in freezing temperatures suggests that the active sites of this sample are less heterogeneous than those of orthoclase.

Freezing curves for the Na/Ca feldspars extend to temperatures where homogeneous freezing sets in, suggesting

that a fraction of these particles, up to 25 % (considering the 800 nm particles in Fig. 3), cannot nucleate ice heterogeneously. In contrast, freezing on the K feldspars (except sanidine) occurs above the homogeneous freezing regime. Potential reasons for why sanidine is an exception to the rest of the K feldspars are discussed in Sect. 5 below. For all K-feldspar samples, the effect of temperature on the frozen fraction has a distinct size dependence where larger particle sizes exhibit a stronger temperature dependence; i.e. the change in frozen fraction with temperature is larger for large particles and smaller for small particles (see Fig. 3). This can be explained by the increased probability of large particles to host at least one exceptionally efficient active site compared to smaller particles with fewer actives sites. Most plagioclase feldspar particles are ineffective heterogeneous INPs or initiate freezing only when temperatures approach homogeneous nucleation temperatures (Harrison et al., 2016).

A comprehensive analysis of how to obtain good immersion freezing parameterizations, including the fitting of the microcline (amazonite) data set shown in this study, can be found in Ickes et al. (2017). Parameterizations for other samples can be found in Niedermeier et al. (2015) and Peckhaus et al. (2016). Compared to the temperature-dependent increase in frozen fraction predicted by a single contact angle representation within classical nucleation theory, the observed temperature dependence is less steep. This feature can be parameterized by, e.g., a contact angle distribution (Ickes et al., 2017). In terms of nucleation sites per surface area (ice-nucleation-active sites – INASs), the number of sites increases by 1 order of magnitude with every additional 2 K of supercooling for most samples (see Fig. A1 in Appendix A). Sanidine and pericline show a weak increase in frozen fraction with decreasing temperature.

### 4.2   Size dependence

The minimum particle size able to trigger heterogeneous ice nucleation above temperatures where droplets freeze homogeneously is investigated within the limits of the experiment (see Sect. 3.1.). The range of homogeneous nucleation temperatures for water droplet sizes of 10–20 µm generated in IMCA was found to be 235–237 K (see Fig. 3). The lower particle threshold size for heterogeneous ice nucleation determines how effectively small inclusions, internally or externally mixed with mineral dust aerosol, will impact ice formation.

Experiments on single immersed, size-selected particles show a strong particle size dependence of ice nucleation ability. The specific slope of the median freezing temperature ($T_{50}$) as a function of geometric surface area varies between samples. Examples are the almost linear scaling of frozen fractions with particle size (surface area) observed for 100–200 nm microcline, 200–400 nm orthoclase and 400–800 nm adularia (see Fig. 5). The two orthoclase (adularia) varieties, which show a low ice nucleation activity for particle sizes

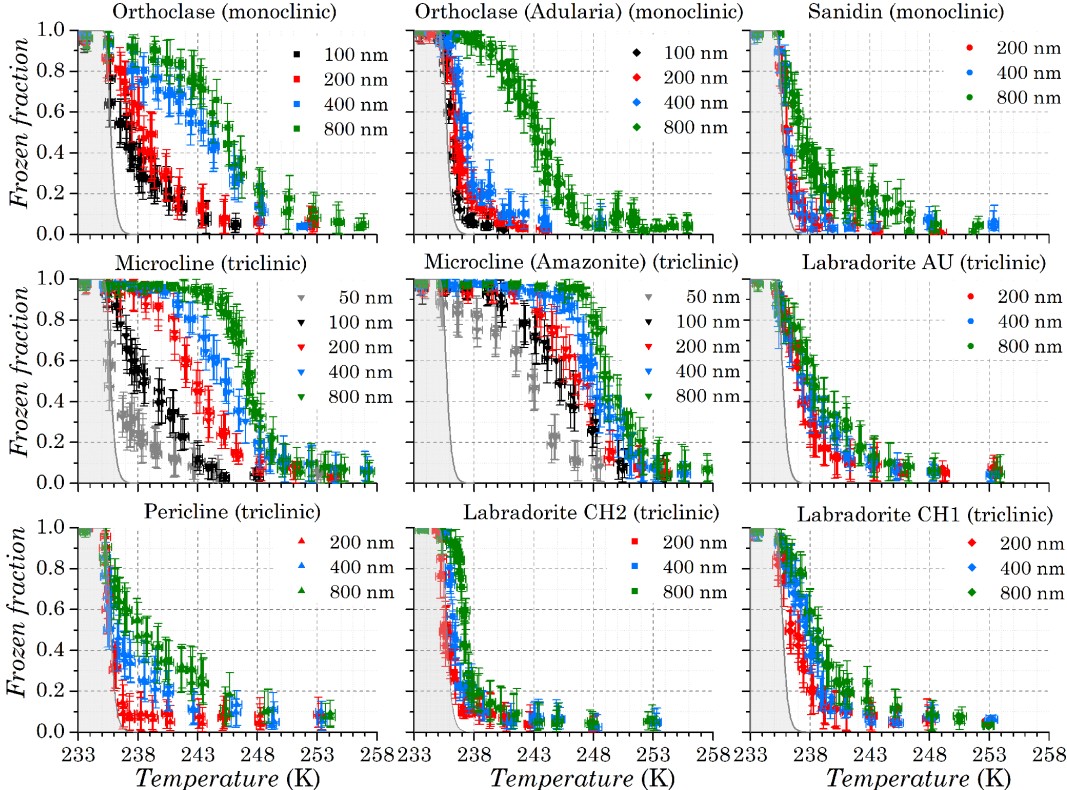

**Figure 3.** Frozen fractions of size-selected feldspar polymorphs. The frozen fraction of water droplets due to homogeneous freezing in the set-up is given by the grey shaded area. Smaller particles (50 and 100 nm) were only tested for the more ice active samples. Horizontal error bars represent the variation in the sample temperature across the aerosol layer in ZINC, and vertical error bars represent uncertainty in the measured frozen fraction due to overlap in depolarization ratio of droplets and ice crystals in the IODE signal (Lüönd et al., 2010). CE3

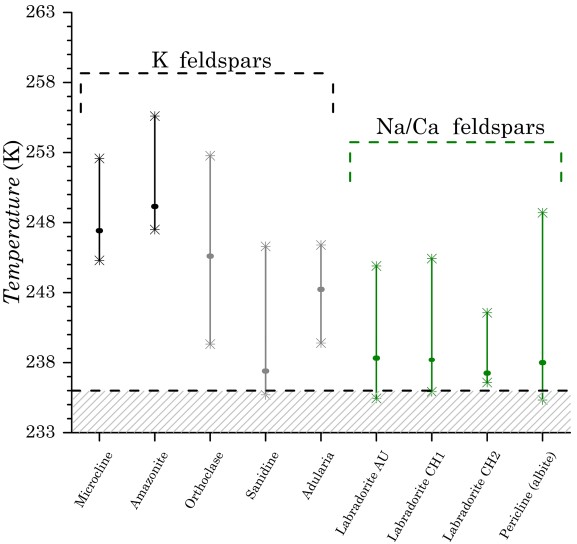

**Figure 4.** Spread in freezing temperatures corresponding to frozen fraction of 0.1 to 1, for 800 nm feldspar particles. Hatched region indicates homogeneous freezing as the dominant mechanism.

smaller than 200 (400) nm are observed to have a stronger increase in activity with surface area compared to the highly active samples (e.g. amazonite). This behaviour could suggest that for larger surface areas than accessible with these experiments, differences among K-feldspar species may become smaller as the $T_{50}$ converge for the K feldspars (Fig. 5). The minimum size triggering immersion freezing is found to be 50 nm microcline particles. In particular, the amazonite sample exhibits a $T_{50}$ of 243 K, which is higher than previously reported values of a variety of dust and clay samples (Lüönd et al., 2010; Welti et al., 2012; Kanji et al., 2013; Augustin-Bauditz et al., 2014; Hartmann et al., 2016). For most of the feldspar samples, including orthoclase and sanidine, particles of 400 or 800 nm are required to observe effective immersion freezing, which contradicts the hypothesis of Atkinson et al. (2013) that generally tiny feldspar inclusions in dust particles will contribute to active ice nucleation. An exception is the microcline (amazonite) sample showing that even particles as small as 50 nm are able to act as INPs, suggesting that the hypothesis could be true for this K-feldspar polymorph. We note that despite the multiply charged particles expected in the 50 nm particle population (see Sect. 3.1),

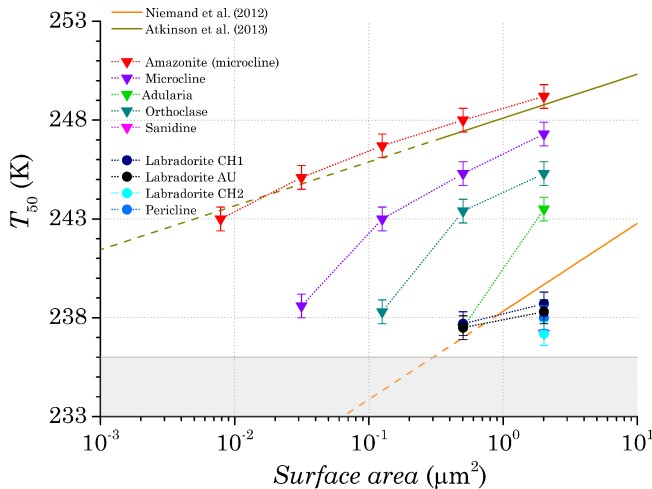

**Figure 5.** Median freezing temperature ($T_{50}$) where 50 % of particles are observed to be frozen as a function of geometric particle surface area. Parameterizations for desert dust from Niemand et al. (2012) TS4 and K feldspar from Atkinson et al. (2013) are included for comparison. Dashed lines indicate extrapolation of the parameterization to lower temperatures. Frozen fractions, where $T_{50}$ was not reached above homogeneous freezing temperatures, have been excluded.

particles of this size substantially contribute to the frozen fractions observed, as shown in Fig. C1 (Appendix C).

## 5 Discussion

The feldspar samples used in this study show a large variability in ice nucleation temperatures. Here, we discuss properties of the feldspar group that could help to explain and categorize this variability. Pruppacher and Klett (1997) list five requirements of an ice-nucleating substance: insolubility, active sites, size, ability to form a chemical bond with water and crystallographic similarities to ice. For the analysis of our experiments, we can deduce the following: (a) the insolubility of feldspar on the timescale of the experiment is reasonable to assume, (b) active sites could be chemical, crystallographic or morphological features discussed in Sect. 5.1., and (c) the particle size effect on ice nucleation in the immersion mode as already discussed in Sect. 4.2. We therefore explore the chemical composition and crystallography of the samples as ice nucleation requirements in the following.

### 5.1 Crystal structure and surface texture

None of the K-feldspar samples exhibits a close lattice match (see Table 2) to hexagonal ice Ih ($a = 4.52\,\text{Å}$ TS5, $c = 7.35\,\text{Å}$) or cubic ice Ic ($a = 6.35\,\text{Å}$), typically found at lower temperatures. There are two crystal structures in the feldspar samples tested: monoclinic and triclinic. While Na/Ca plagioclases are all triclinic, the K-feldspar sanidine and orthoclase

have a monoclinic crystal lattice. Microcline is a triclinic K feldspar. Triclinic microcline forms by the transformation of monoclinic K feldspar upon cooling (see, e.g., Waldron et al., 1993) or, as suggested by Collins and Collins (1998), TS6 by K replacement of Na/Ca feldspar. The lattice angles of monoclinic and triclinic K-feldspar polymorphs are summarized in Table 2. It can be seen that structural differences among feldspars are small even when comparing triclinic to monoclinic lattice angles. Mason and Maybank (1958) reported experiments on triclinic microcline and monoclinic orthoclase. While microcline was found to be active at $-9.5\,°C$, their orthoclase sample was inactive above $-18\,°C$. The data shown in Fig. 3 support a difference in ice nucleation temperature of these K feldspars, with microcline forming ice at higher temperatures than the monoclinic orthoclase, although the absolute temperatures differ from Mayson and Maybank (1958) TS7 because of the single-particle nature of our studies. In addition, the least efficient K-feldspar sample sanidine is monoclinic. The hypothesis that within the K-feldspar polymorph, a triclinic crystal structure is a more active ice nucleation property was reported by Augustin-Bauditz et al. (2014), based on their indirect observation that mixed dust samples containing orthoclase were less ice active than a microcline sample. However, given that orthoclase is much more ice active than sanidine (Fig. 3) and both have a monoclinic structure, their difference in ice activity cannot be explained by the crystal lattice alone. Another difference between monoclinic and triclinic K feldspars is the order or disorder of $Al^{3+}$ and $Si^{4+}$ in the tetrahedral sites building the crystal structure. Monoclinic sanidine retains a disordered crystal structure (although often forms defect-free crystals), while microcline transforms to an ordered crystal. Orthoclase has intermediate disorder in this regard. The polymorphs of K feldspar are distinguished by the randomness of this order distribution, which depends on the temperature and cooling rate during rock formation. The sequence of feldspar melting temperatures given in Lutgens et al. (2014) is Ca feldspar > Na/Ca feldspar > Na feldspar > K feldspar (sanidine > orthoclase > microcline). While Ca feldspar is completely ordered, Na feldspar can have varying degrees of order and Na/Ca feldspars are disordered. The melting points of K feldspar correspond to the ascending order of the observed $T_{50}$ and the degree of order–disorder, indicating that ordered feldspar crystals are more active INPs than disordered crystals. However, since the crystal structures do not match that of ice, it is not possible to state what the reason for such a correlation could be.

The surface texture of the nine samples was investigated by scanning electron microscopy (SEM). Representative images are shown in Fig. 6. Steps, lamellae and clustering are prominent features. Smaller particle sizes appear smoother and rounder. Due to the triclinic structure of Na/Ca feldspar, twinning (i.e. symmetrical intergrowth, phase exosolution or Na–K exchange between feldspar phases) with triclinic microcline is common (David et al., 1995; Parsons et al., 2015;

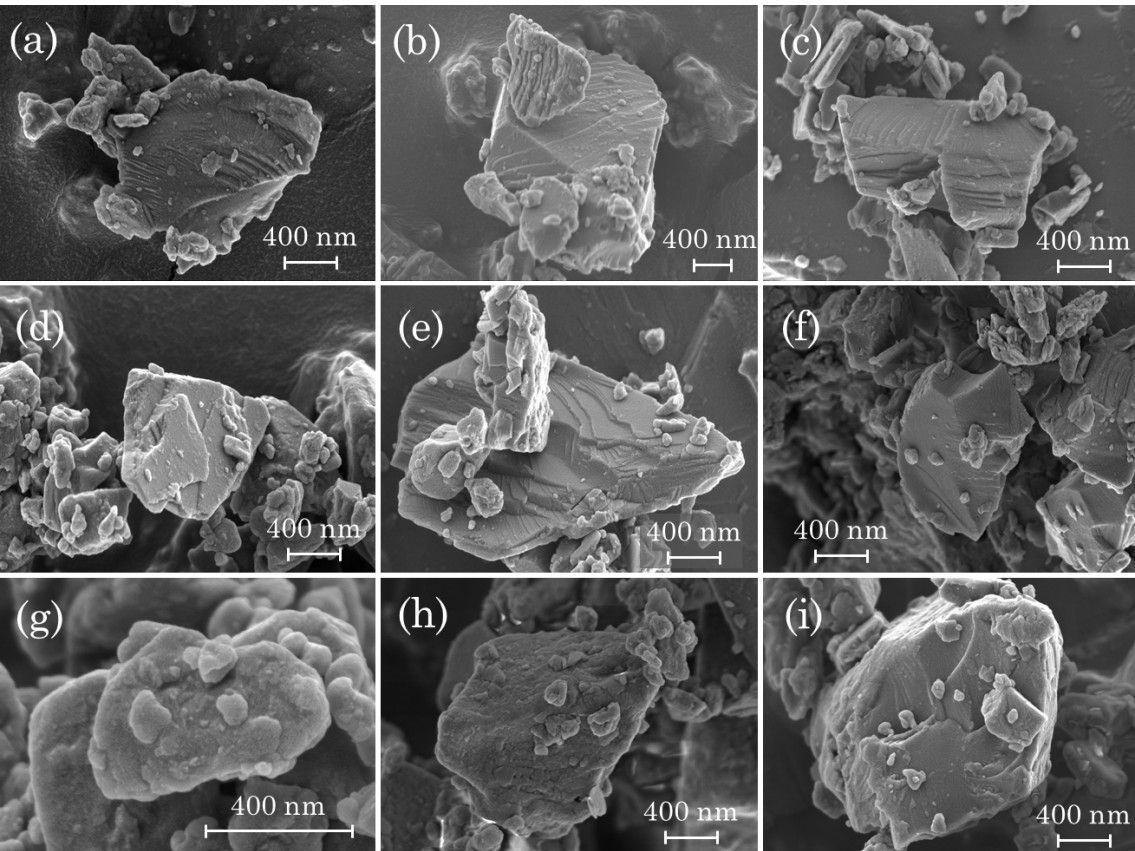

**Figure 6.** Representative scanning electron microscope images of **(a)** orthoclase, **(b)** adularia, **(c)** sanidine, **(d)** microcline, **(e)** amazonite, **(f)** labradorite AU, **(g)** pericline, **(h)** labradorite CH2 and **(i)** labradorite CH1. For reference, the 400 nm scales are indicated in the figures. The lamellae (perthitic) structure can be seen for the K feldspars **(a–e)** and to a lesser extent for the plagioclase feldspars **(f–i)** but is almost absent for pericline **(g)** and labradorite CH2 **(h)**. Images taken using a LEO 1530 Gemini SEM. Dust samples were applied to adhesive, carbon conductive tapes and sputter-coated by a thin platinum layer. High-resolution secondary electron (in-lens SE CE4 detector) images were acquired at 2 kV.

**Table 2.** Lattice parameters for K-feldspar polymorphs from Hovis (1986) and Ostrooumov and Banerjee (2005).

| Feldspar type | a (Å) | b (Å) | c (Å) | $\alpha$ (deg) | $\beta$ (deg) | $\gamma$ (deg) |
|---|---|---|---|---|---|---|
| Microcline | 8.59 | 12.97 | 7.22 | 90.6 | 116 | 87.8 |
| Amazonite | 8.59 | 12.97 | 7.22 | 90.9 | 116 | 87.6 |
| Orthoclase | 8.60 | 13.00 | 7.20 | 90 | 116 | 90 |
| Adularia | 8.60 | 12.97 | 7.21 | 90 | 116 | 90 |
| Sanidine | 8.60 | 13.02 | 7.18 | 90 | 116 | 90 |

Whale et al., 2017). However, given that the $\alpha$ and $\gamma$ crystallographic angles are very close to 90° (see Table 2), it is difficult to detect twinned structures in SEM images since the surface would need to be slightly tilted (Lee et al., 1998, and references therein). The surface sites where transitioning between microcline and plagioclase occurs has been proposed to be a topological feature critical for ice nucleation

(Whale et al., 2017). All K feldspars exhibit perthitic structures (lamellae), whereas on the plagioclase feldspars this feature is less prominent or missing (Whale et al., 2017). In Fig. 6, the layering from the lamellae is clearly visible for panels (a)–(e) (K feldspars), whereas for panels (f)–(i) the layering is less visible and even absent for panels (g) and (h) (pericline and labradorite CH2). It has been reported that K feldspar contains macropores of 0.1–1 µm diameter (David et al., 1995; Lee et al., 1995). Hodson (1998) found that the contribution of internal surface structures such as pores to the total particle surface increases with decreasing particle size for some K feldspars (e.g. sanidine), whereas no particle size relation to pore surface was found for other samples such as microcline. For the current study the internal surface area is not of interest since in comparison to the outer surface in contact with the water droplet, ice formation is depressed in narrow confinements (Marcolli, 2014). Therefore, using the particle surface area available to nitrogen adsorption (BET CE5 surface) as a measure of surface area might overestimate the available surface for immersion freezing nu-

cleation. Geometric surface area assuming a spherical shape is used instead (Fig. 5). In none of the SEM images, intact pores are visible. This is possibly because of the Pt coating prior to imaging. Also, the grinding process applied to the samples may have eliminated this feature, as cavities and macropores of this size present favoured sites of fracture. Lamellae could stem from split pores. Before milling, the K-feldspar samples used here were translucent to opaque, indicating that they reacted pervasively with a fluid causing exsolution, and thus the perthitic structure is expected (Parsons et al., 2015). All samples were subject to the same preparation (grinding) procedure. The assumption that physical properties (e.g. hardness) are comparable among the tested feldspar species implies that the same degree of artificial surface features are introduced to all samples. The vast difference in ice nucleation efficiency between the feldspars suggests that any grinding artefacts (such as fractures or exposing high-energy surfaces) did not create ice-nucleating properties, as otherwise all samples would have more comparable ice nucleation activities. Zolles et al. (2015) tested the effect of milling on a sample of microcline, albite and andesine and found none or only a minor increase of 1 K on the median freezing temperature. We note that this is not the case for all minerals as a large influence of milling has been noted for haematite (Hiranuma et al., 2014).

## 5.2 Chemical composition

The individual samples have been analysed by XRF spectroscopy for their bulk chemical composition and trace elements (see Table 3). Using actively ground samples from a homogeneous base material should preclude large variations in chemical composition with particle size for the samples investigated here (see INAS densities in Appendix A). A statistical classification of the XRF results using random forest analysis (Liaw and Wiener, 2002) to identify the important compositional predictors of $T_{50}$ revealed Pb, Rb and Sr to be most relevant. Pb and Rb are highly positively correlated ($r = 0.91$), whereas the concentration of Sr in the samples shows a negative correlation to Pb ($r = -0.66$) and Rb ($r = -0.51$). The best linear correlation between a single compound and the ice nucleation activity based on $T_{50}$ was found for the Pb content in the feldspar samples (Fig. 7). However, there is an exception with using Pb as the single compound predictor since the microcline sample has a lower lead content than the orthoclase but a higher $T_{50}$. Cziczo et al. (2009) have shown that the presence of Pb could increase the ice nucleation activity of dust (clay) particles. However, high Pb content alone is insufficient to explain the ice nucleation activity of a mineral dust sample in general. As an example, the ice nucleation activity of a weathering product of feldspars (kaolinite) does not appear to depend on its Pb content. XRF analysis of the kaolinite samples used in the experiments reported in Welti et al. (2012), which had a lower ice nucleation activity than microcline, revealed a

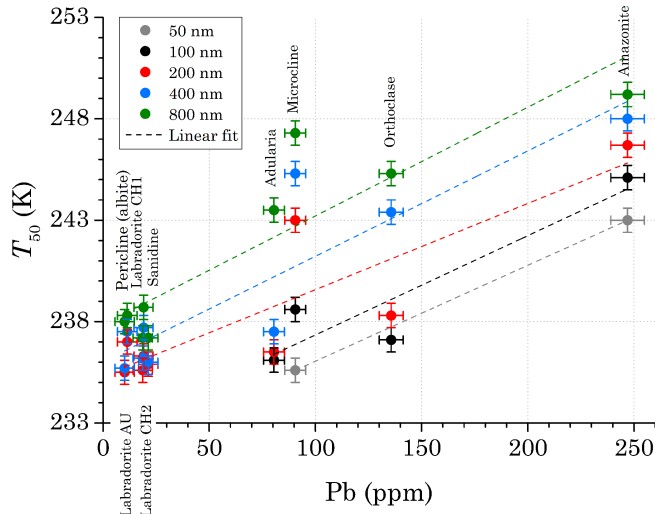

**Figure 7.** Particle-size-dependent median freezing temperature ($T_{50}$) as a function of lead (Pb) content. Particle size is indicated by the coloured legend.

~ 10-fold higher Pb content (2888.4 ppm) in comparison to, e.g., the amazonite sample (247.0 ppm). The high Pb content measured in kaolinite could be a result of its high adsorption capacity for Pb (Jiang et al., 2010) and can be introduced at any point after the weathering from feldspar to kaolinite. Based on these arguments and that the correlation of Pb content to $T_{50}$ is not monotonic (e.g. lower Pb content in microcline than orthoclase; Fig. 7) we focus on other predictors suggested by the random forest analysis.

The strongest chemical predictor for $T_{50}$ was found to be a high Rb/Sr ratio. Contrary to the Pb content, a higher Rb/Sr monotonically corresponds to a higher $T_{50}$ for all samples and particle sizes, shown in Fig. 8. Initially each feldspar has a specific Rb/Sr partitioning influenced by its K content, the melt temperature from which it crystallized and its melt composition. If it is subsequently changed, the Rb/Sr ratio can be an indication of mineralization processes (Plimer and Elliott, 1979). Due to their ionic radii, electronegativity and ionization potentials, $Rb^+$ replaces $K^+$ while $Sr^{2+}$ is incorporated in the feldspar crystal instead of $Ca^{2+}$. The Rb/Sr ratio is indicative of the position of feldspar on the ternary phase diagram (Fig. 1) and the absolute concentrations of trace elements depends on the cooling or crystallization process and abundance of trace elements in particular magmatic fluid (Parsons et al., 2009). The replacement of K by Rb results in a lower microcline polymorph and a higher Rb/Sr ratio, correlating to higher ice nucleation activity. This suggests that the observed correlation reflects a possible relationship between ice nucleation activity and feldspar microstructure. We note that a monotonic correlation between $T_{50}$ and a Pb/Nd ratio can also be constructed, but no interpretation of the implication of such a ratio could be found in the literature and neither was Nd suggested as a predictor by the ran-

**Table 3.** Chemical analysis from XRF of the different feldspar samples used in this study. N/D stands for not detectable but could be above zero. A zero implies reliable absence of a compound.

| | Plagioclase feldspar | | | | Potassium feldspar | | | | |
|---|---|---|---|---|---|---|---|---|---|
| | Labradorite CH1 | Labradorite CH2 | Labradorite AU | Pericline (albite) | Adularia | Sanidine | Orthoclase | Microcline | Amazonite |
| Compound | | | | | wt % | | | | |
| $SiO_2$ | 55.05 | 56.809 | 60.429 | 73.471 | 64.035 | 64.51 | 64.841 | 64.794 | 65.357 |
| $TiO_2$ | 0.095 | 0.117 | 0.192 | 0.005 | 0.008 | 0.017 | 0.06 | 0.005 | 0.005 |
| $Al_2O_3$ | 27.915 | 26.404 | 23.366 | 16.524 | 18.87 | 18.753 | 18.742 | 18.618 | 18.490 |
| $Fe_2O_3$ | 0.254 | 0.39 | 0.554 | 0.027 | 0.032 | 0.177 | 0.268 | 0.044 | 0.041 |
| MnO | 0.006 | 0.006 | 0.009 | 0.002 | 0.001 | 0.002 | 0.005 | 0.004 | 0.003 |
| MgO | 0.007 | 0.087 | 0.216 | 0 | 0 | 0 | 0.11 | 0 | 0 |
| CaO | 10.415 | 8.224 | 6.702 | 0.055 | 0 | 0.041 | 0.392 | 0.064 | 0.019 |
| $Na_2O$ | 5.148 | 6.591 | 5.32 | 9.918 | 0.978 | 1.719 | 2.337 | 1.836 | 2.551 |
| $K_2O$ | 0.587 | 0.21 | 1.004 | 0.014 | 14.749 | 13.94 | 12.176 | 13.965 | 12.588 |
| $P_2O_5$ | 0.017 | 0.05 | 0.035 | 0.007 | 0.004 | 0.035 | 0.036 | 0.026 | 0.008 |
| $Cr_2O_3$ | 0.001 | 0.001 | 0.007 | N/D | N/D | N/D | 0.001 | N/D | 0.001 |
| LOI | 0.4 | 0.86 | 2.006 | 0.526 | 0.407 | 0.293 | 0.62 | 0.58 | 0.384 |
| Total | 99.894 | 99.75 | 99.84 | 100.549 | 99.084 | 99.487 | 99.587 | 99.936 | 99.447 |
| Element | | | | | Traces (ppm) | | | | |
| Rb | 0.9 | 0 | 14.6 | 3 | 426.1 | 208.8 | 459 | 460.9 | 4566.8 |
| Ba | 161.2 | 225.9 | 180.7 | 0 | 10591.7 | 4883.4 | 1812.6 | 198.6 | 324.8 |
| Sr | 991.9 | 1567.5 | 746.4 | 274.2 | 720.6 | 1236.6 | 295.2 | 42.9 | 30.7 |
| Nb | 1.3 | 2.8 | 3.7 | 3.2 | 1.7 | 3.3 | 2.9 | 1.8 | 8.9 |
| Zr | 0 | 0 | 30.4 | 0 | 0 | 0 | 34.6 | 0.7 | 2.2 |
| Hf | 5.2 | 6.4 | 5.1 | 2.4 | 3.8 | 5.6 | 3.8 | 2.8 | 0.0 |
| Y | 2.8 | 2.9 | 6 | 0.3 | 0 | 0.4 | 0 | 0 | 0.0 |
| Ga | 29.7 | 24.4 | 25.7 | 15.3 | 19 | 27.4 | 26.8 | 24.2 | 91.6 |
| Zn | 10.6 | 7.8 | 13.4 | 2.7 | 5.2 | 7.2 | 11.6 | 5.8 | 10.4 |
| Cu | 6.8 | 20.6 | 121.1 | 0 | 33 | 94.5 | 75.4 | 3.9 | 1515.4 |
| Ni | 0.8 | 3.8 | 3.3 | 0 | 0.7 | 0 | 0.9 | 0 | 0.4 |
| Co | 21.3 | 40.6 | 1.3 | 43.7 | 21.6 | 14.5 | 16.2 | 32.5 | 13.6 |
| Cr | 5.1 | 8.8 | 51.7 | 2.2 | 2.1 | 2.8 | 8.9 | 2.6 | 5.7 |
| V | 4.3 | 4.4 | 13.3 | 1.8 | 3.8 | 3.1 | 3.6 | 1.9 | 10.1 |
| Sc | 0 | 0 | 1.2 | 0 | 0 | 0 | 3 | 0.6 | 1.8 |
| La | 0 | 1.3 | 7.9 | 7.4 | 0 | 0 | 13.8 | 3.1 | 82.2 |
| Ce | 0 | 0 | 22.8 | 0 | 64.7 | 22.5 | 32 | 0 | 3.3 |
| Nd | 9 | 6.4 | 16.6 | 8.1 | 32.4 | 17 | 19.6 | 8.4 | 8.7 |
| Pb | 19.1 | 18.7 | 11.4 | 10.2 | 80.5 | 21.3 | 135.7 | 90.5 | 247.0 |
| Th | 0 | 0 | 2.9 | 0.8 | 0 | 0 | 7.7 | 0.2 | 0.0 |
| U | 0 | 0 | 0 | 0 | 0 | 0 | 0.5 | 0 | 2.1 |

dom forest analysis. Zolles et al. (2015) proposed an explanation of K-feldspar ice nucleation ability based on its surface chemistry. They argue that depending on the chaotrope (order-breaking) or kosmotrope (order-making) characteristic of feldspar surface cations which are present in the water surface surrounding the particle, they can enhance or inhibit the structuring of water and therefore the formation of the ice phase. The ions with the higher charge densities ($Ca^{2+}$, $Na^+$, $Sr^{2+}$) tend to disturb the structuring of water (increase entropy) while those with a lower charge density ($K^+$, $Rb^+$) have a promoting effect (decrease entropy). Following this argument, an increased $Rb^+$ concentration combined with a decreased $Sr^{2+}$ content results in a more kosmotropic and less chaotropic surface property for the ordering of water into ice. The increased ice nucleation activity with an increasing Rb/Sr ratio therefore corresponds to a higher ordering and thus supports the hypothesis of Zolles et al. (2015). It would be necessary to quantify the concentration of cations required to influence water ordering to determine how influential $Rb^+$ and $Sr^{2+}$ are for ice nucleation given their trace elemental composition.

The observed discontinuities in freezing temperature with increasing particle size for adularia and orthoclase (see Fig. 3) indicate that chemical composition alone cannot explain the ice nucleation potential of all feldspar species. That the strong size dependence is observed for both orthoclase

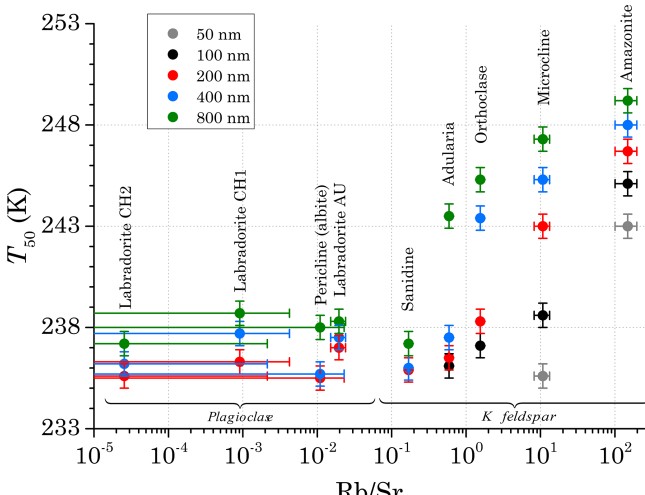

**Figure 8.** Particle-size-dependent median freezing temperature ($T_{50}$) as a function of the Rb (rubidium) / Sr (strontium) ratio. Particle size is indicated by the coloured legend.

varieties (intermediate ordered crystal structures) but not for disordered sanidine indicates that a large enough, sufficiently ordered surface area is needed to initiate efficient ice nucleation. This interpretation corroborates the suggestion of Whale et al. (2017) that some perthitic structure (microtexture) associated with the strain between K- and Na-rich regions on the samples benefits ice nucleation, and these can range from the nanometre to the millimetre scale, varying amongst feldspar samples. The probability of such features inevitably scales with surface area and could explain the requirement of a critical particle size (surface area) in order to exhibit ice nucleation.

Natural weathering producing small feldspar particles commonly includes contact to acidic environments, causing the removal of alkalis and aluminium and the formation of a silica-rich skin. Freely exposed feldspars are reported to be populated by lichens, which can alter naturally occurring feldspar in an infinite number of ways (Smith, 1998). The surface properties (specific area, porosity, molecules on the surface) of freshly ground feldspar are therefore expected to be different from aged particles (Mangan et al., 2017). However, bulk composition and crystal structure can be assumed to be representative of natural feldspar. Additionally, a majority of airborne feldspar particles could be coming from phenocrysts (inclusions of feldspar in, e.g., a granite matrix; Lee et al., 1995), where the trace element content could be different. The ageing of the microcline in concentrated solutions did not impair ice nucleation activity permanently in the case of near-neutral solutions; however, exposure to highly acidic or alkaline solutions damaged the surface, inhibiting or even destroying the ice nucleation ability (Kumar et al., 2018). We note that the samples used in this study could have different ice nucleation properties from feldspar found

in natural airborne dust because of the natural weather elements such as cold, low-pH conditions, resulting in a decrease in ice nucleation activity. On the other hand, high relative humidity with high-$NH_3$/$NH_4^+$ conditions could enhance ice nucleation (Kumar et al., 2018). Feldspars of the same classification but from a different location could differ in immersion mode freezing properties given that feldspars can exhibit broadly similar chemistry and structure but at the atomic level will almost certainly differ in their intracrystal defects as well as in their trace element chemistry (Lee et al., 1998; Parsons et al., 2015) as shown in Figs. 5 and 6.

## 6 Conclusions

The immersion ice nucleation properties of size-selected K- and Na/Ca-feldspar particles have been investigated in the temperature range 235–258 K. A pronounced sample-dependent effect of particle size on immersion freezing activity is observed. The analysis of composition, lattice structure and crystallographic properties of the samples in correlation to their ice nucleation behaviour suggests two indicators for the ice nucleation activity of the feldspar samples investigated here, namely, high crystal order and the abundance of certain trace elements.

The tested K feldspar (microcline) with high crystal order is found to exhibit a higher ice nucleation activity compared to chemically similar intermediate or highly disordered K-feldspar polymorphs (orthoclase, sanidine). The lower the K-feldspar crystallization temperature, the higher the order in crystal structure, favouring the formation of microcline. Accordingly, the Na/Ca-feldspar samples investigated in this study crystallized at higher temperatures and exhibit lower ice nucleation activities. Crystal structure (triclinic or monoclinic) as a template for ice nucleation is found to be not important, as was also concluded by molecular model simulations (Pedevilla et al., 2016). The best predictor of the median freezing temperature ($T_{50}$) is found to be the sample Rb/Sr ratio, determined by XRF. More ice-nucleation-active samples show higher Rb/Sr ratios; for example, the most active microcline sample has an Rb/Sr ratio that is an order of magnitude higher than that of the second-best microcline. Therefore, Rb/Sr ratios could serve as tracer for highly ice-nucleation-active feldspar particles or even help to differentiate sources of feldspars acting as INPs at specific temperatures.

Microcline and plagioclase feldspars have been found in natural dust surface samples from deserts (Kaufmann et al., 2016) and to a lesser extent (but still significant amounts) in airborne dust samples (Boose et al., 2016). Particle size, wind speed and turbulence determine atmospheric transport and the spread of dust particles (Mahowald et al., 2014). The size-dependent immersion freezing activity therefore affects the temperature range at which certain feldspar particles contribute to cloud glaciation. Size-dependent measure-

ments indicate that for the two orthoclase samples (orthoclase, adularia), ice nucleation requires active sites present on 400–800 nm sized particles but not on the smaller size fraction (100–200 nm in diameter). The size-dependent results further suggest that even small microcline particles or inclusions in clay minerals could contribute substantially to the ice activity, given the smaller size fraction (50–100 nm) of microcline that formed ice above homogeneous freezing temperatures. A possible caveat is that atmospheric ageing can change the surface of the microcline, thus rendering its ice activity dependent on the type of chemical ageing experienced. We conclude that the larger particle size required to trigger ice formation at temperatures above homogeneous freezing for other K-feldspar samples would prevent their inclusions within mineral dust particles to increase the ice nucleation activity of airborne mineral dusts. The experiments of the present study suggest that sanidine (most abundant K feldspar in volcanic ash) is not an active INP.

We note that the point of origin and sampling location of the feldspars used here could limit the specific ice nucleation activity to the samples presented here because even similar polymorphs sampled in different geological landscapes are known to have different topographical features and microtextures (Lee et al., 2007). However, the results from this work demonstrate the variation in ice nucleation activity amongst one group of minerals in the submicron particle range.

Additional experiments with volcanic K feldspars (sanidine varieties) might help to explain the observed differences in the ice nucleation efficiency of ashes from different volcanoes (e.g. Mangan et al., 2017) and CE7 could benefit model-based studies, which forecasts aerosol plumes after volcanic eruptions to guide policy for aviation safety. Ice nucleation investigations with a larger number of samples of one specific feldspar polymorph, from different source regions, could help to generalize the finding that a trace elemental fingerprint can be used to predict the ice nucleation activity of feldspar samples.

*Data availability.* The data presented in this publication are available at https://doi.org/10.3929/ethz-b-000350978 (Kanji et al., 2019). CE8

Please note the remarks at the end of the manuscript.

## Appendix A: Ice-nucleation-active site (INAS) densities of feldspar polymorph samples

Based on the experimental data shown in Fig. 3 (Sect. 4), INAS density ($n_s$) scales the frozen fraction (FF) by the particle surface area according to

$$n_s = \frac{\ln(1 - \text{FF})}{A},\qquad(\text{A1})$$

where $A$ denotes the geometric surface area, calculated as $A = 4\pi \left(\frac{d}{2}\right)^2$, with $d$ being the selected mobility diameter. We have determined these for the all the feldspar samples as shown in Fig. A1.

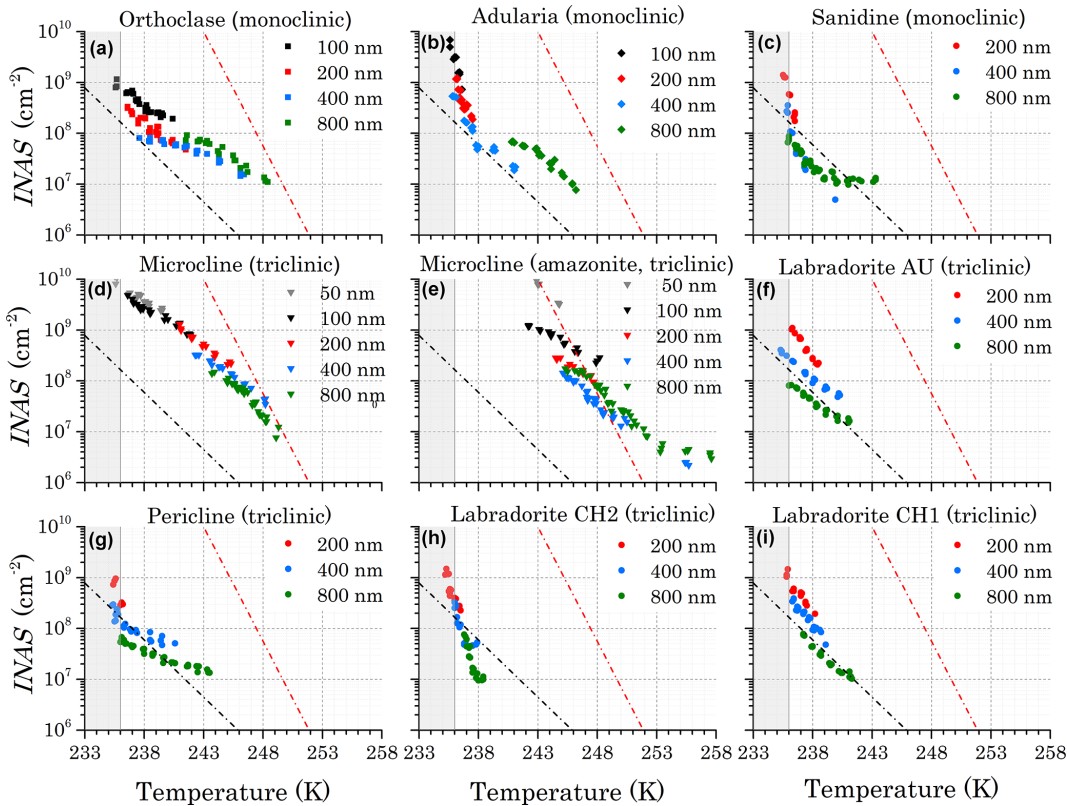

**Figure A1.** INAS densities of nine feldspar samples as a function of temperature and size corresponding to observed frozen fractions as shown in Fig. 3. The upper (80 %–100 %) and lower (0 %–20 %) frozen fractions have been omitted to exclude saturation errors from the detection.

## Appendix B:  Sample mineralogy determined by X-ray diffraction (XRD) analysis

The composition of samples investigated here were additionally determined by Rietveld refinement of powder XRD pattern. We note that this method is subjective to the fitting procedure to some degree. Results are given in Table B1.

**Table B1.** Mineralogical composition (mass %) of feldspar samples as determined by powdered XRD using the Rietveld refinement method.

| Sample | Orthoclase | Adularia | Sanidine | Microcline | Plagioclase | Quartz | Others |
|---|---|---|---|---|---|---|---|
| Orthoclase | 70 | | | | 15 | 5 | 10 |
| Adularia | | 100 | | | | | |
| Sanidine | | | 100 | | | | |
| Microcline | | | | 90 | 10 | | |
| Microcline (amazonite) | | | | 77 | 22 | 1 | |
| Labradorite (AU) | | | | | 59 | 12 | 29 |
| Pericline | | | | | 84 | 16 | |
| Labradorite CH2 | | | | | 100 | | |
| Labradorite CH1 | | | | | 100 | | |

## Appendix C: Contributions of singly, doubly and triply charged particles to the 50 nm microcline samples

Multiple charge contribution to measured frozen fractions of size-selected particles can be estimated from the initial particle size distribution, the sizes of the singly, doubly and triply charged particles, and the fraction of total particle concentration that carries one, two or three charges (Wiedensohler et al., 1986). The calculation is reproduced for 50 nm microcline below.

| Charge | Mobility diameter | Fraction of total particle concentration carrying this charge | Percentage of total selected particles |
|---|---|---|---|
| + | 50 nm | 0.17 | 63 |
| ++ | 73 nm | 0.016 | 30 |
| +++ | 91 nm | 0.0012 | 7 |

Frozen fraction (FF) can be calculated according to Eq. (A1) using INAS density or by classical nucleation theory:

$$FF = 1 - \exp(-J(T) \cdot A \cdot t), \qquad (C1)$$

With $J(T)$ being the nucleation rate, $A$ the particle surface area and $t$ time. Using the assumption that the temperature-dependent nucleation rate $J(T)$ or INAS density are sample specific, not a particle-size-dependent property, and that ice nucleation activity scales with particle size, the contribution of each particle size to the total frozen fraction (shown in Fig. C1) can be calculated:

$$\text{Contribution}_{50\,\text{nm}} = \frac{0.63 \cdot \text{FF}_{50\,\text{nm}}}{\text{FF}_{\text{tot}}}, \qquad (C2)$$

$$\text{Contribution}_{71\,\text{nm}} = \frac{0.30 \cdot \text{FF}_{73\,\text{nm}}}{\text{FF}_{\text{tot}}}, \qquad (C3)$$

$$\text{Contribution}_{93\,\text{nm}} = \frac{0.07 \cdot \text{FF}_{91\,\text{nm}}}{\text{FF}_{\text{tot}}}. \qquad (C4)$$

The result is insensitive to the nucleation rate $J(T)$ or INAS density used to calculate the frozen fractions, but the same $J(T)$ or INAS density must be used for all sizes. From Fig. C1, it can be seen that even for the lowest frozen fractions more than 40 % of the particles are indeed 50 nm, whereas for frozen fractions larger than 0.5 TS8, 50 % or more particles are 50 nm in mobility diameter.

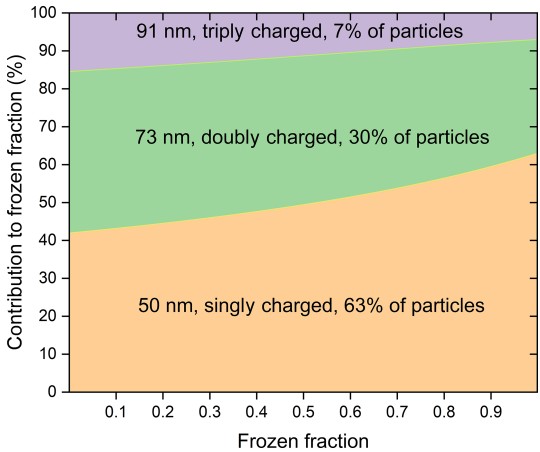

**Figure C1.** Contribution of singly, doubly and multiply charged particles to the frozen fraction of 50 nm microcline and amazonite samples.

*Author contributions.* AW conducted the experiments and analysed the data. AW and ZAK interpreted the data. AW and ZAK wrote the paper with comments from UL. ZAK and UL supervised the project.

*Competing interests.* The authors declare that they have no conflict of interest.

*Acknowledgements.* Zamin A. Kanji and André Welti acknowledge funding from the Swiss National Science Foundation (grant 200020_150169/1). We acknowledge TS9 H. Wydler for technical support. The authors acknowledge Y. Boose for helpful discussions.

*Financial support.* This research has been supported by the Swiss National Science Foundation (grant no. 200020_150169/1). TS10

*Review statement.* This paper was edited by Ottmar Möhler and reviewed by Alexey Kiselev and one anonymous referee.

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

## Remarks from the language copy-editor

## Remarks from the typesetter