# Peer review of "Ice nucleation properties of K-feldspar polymorphs and plagioclase feldspars"

_Atmospheric Chemistry and Physics, 2018_

## Referee Comment (RC1) · Anonymous Referee #1 · 16 Jan 2019

**Review of "Ice nucleation properties of K-feldspar polymorphs and plagioclase feldspars" by André Welti and co-authors for Atmospheric Chemistry and Physics**

General comments:

Welti et al. present a study about the immersion freezing behavior of a variety of different feldspar samples which builds on recent investigations by Augustin-Bauditz et al. (2014), Peckhaus et al. (2016), Harrison et al. (2016), and Whale et al. (2017). The samples were chosen carefully to provide a variety of crystal structures, chemical compositions, and ordering of the crystal lattice. They include five polymorphs of K-feldspar and four plagioclase feldspars. The immersion freezing experiments were performed with droplets containing single, size-selected particles and care was taken to minimize the amount of multiply-charged particles in the sample aerosol. Furthermore, the study includes X-ray fluorescence measurements giving information about bulk chemical composition and scanning electron microscopy images of particle morphology.

What differentiates the present paper from the earlier studies is the discussion of the effect of particle size and the degree of order in the crystal network on the ice nucleation efficiency of the samples. The authors' conclusions concerning these factors are generally comprehensible and well substantiated by the presented results. However, in some cases, which are pointed out in the specific comments, I am missing a more precise explanation. The figures are mostly clear, but I would like to suggest some alternatives for presenting the data (see specific comments). Language-wise, the paper is concisely written but some minor adjustments would increase readability (see technical corrections).

Overall, the paper is interesting, understandable, and fits within the scope of Atmospheric Chemistry and Physics. I recommend publication after minor editing.

Specific comments:

1) Size dependence of ice nucleation behavior

The authors often refer to the "pronounced size dependence of ice nucleation activity" as if this is a rarely observed feature. Normally, the ice nucleation behavior scales with the surface area of the immersed particles, meaning that the efficiency increases with increasing particle size. The authors should clarify to what extent the size dependent ice nucleation behavior of their samples deviates from the standard. In my opinion, this is best done by calculating the ice nucleation active surface site density $n_s(T)$ for differently sized particles. In contrast to the chosen $T_{50}$ approach, this method would have the benefit of providing an overview over the whole investigated temperature range. I hence suggest to replace Fig. 5 with a multi-panel figure (like Fig. 3) showing $n_s(T)$ of the different particle sizes for all investigated samples (see, e.g., Fig. 5 in Hartmann et al., 2016).

Furthermore, the authors state that "microcline exhibited immersion freezing even for 50 nm particles" whereas for orthoclase "ice nucleation requires active sites present on 400-800 nm sized particles" and relate this observation directly to the effect of these particles on atmospheric ice nucleation. Concerning the potential of these species as atmospheric INP, one must always combine their efficiency and their abundance. Larger orthoclase particles might be needed to trigger ice nucleation as efficiently as smaller microcline particles, but maybe many more orthoclase particles are emitted into the atmosphere? Besides, the fact that ice nucleation was not observed for 100 or 200 nm orthoclase particles is related to the detection limit of the instrument. If the authors had investigated more droplets, they would eventually have observed ice nucleation triggered by the small orthoclase particles. This should be made clear in the manuscript.

2) Difference to earlier studies

This refers to P3L14-18, where I think the authors should clarify the innovation of their study more. Like this, it sounds as if they might expect an effect of methodology on the results, as the other studies

were performed with droplets containing numerous particles each. Please state that by using single particles, you are focusing on a different temperature range than the other studies (except Augustin-Bauditz et al., 2014).

3) Multiply-charged particles

The authors should be more precise concerning the amount of multiply-charged particles in the cases where the CPMA was not used. This issue could be addressed by including actually measured size distributions in Fig. 2 instead of the schematic ones. Alternatively, the authors could include the following statement on P4L28-29: "The use of the CPMA for the selection of larger particles (400 nm, 800 nm) was not necessary as the fraction of larger particles was reduced to … % by the cyclones and the impactor upstream of the DMA."

4) Figure 4

I see more benefit from one figure showing *FF* over *T* for 800 nm particles of all samples. This would be more suited for comparing the ice nucleation signatures of the different feldspars than just showing the range in which freezing occurred. Error bars could be omitted (because they are already shown in Fig. 3) and symbol size reduced for clarity.

Technical corrections:

P1L9: Replace "Na/Ca-rich feldspar" with plagioclase to be consistent with the title. The composition is explained below anyhow.
P1L11: Replace "are" with "were" in "Samples are selected…".
P1L18-20: This sentence would benefit from being split into two.
P1L24-25: Either omit the "s" at "temperatures" or at "depends".
P1L29: There is also contact freezing in which the contact causes nucleation, not an immersed particle.
P2L12: Within a sentence "e.g." should be preceded by a comma. A comma should also follow in case you are using American English. This also applies to "i.e.". Generally, check your manuscript for consistency with either British or American English. E.g., see "favouring" on P11L10 or "generalise" and "analysed" on P12L8 and L13.
P2L13: Less efficient in comparison to which other species?
P2L14: Change "for example" to "e.g.".
P2L22: Capital "X" in "x-ray".
P3L21: Mention that XRF is a bulk, not a single particle technique.
P3L28: Omit comma following "polymorphism" and add "s" to "occur".
P3L30-31: Add comma behind "sanidine". Be consistent using either "temperature" or "temperatures".
P4L1-2: I suggest to remove the brackets and structure the sentence as follows: "sanidine in volcanic and very high-temperature metamorphic rocks, orthoclase in … rocks and microcline in … rocks."
P4L2: "feldspar": This should be plural.
P4L3: Why is Table 3 referred to before Table 2 is mentioned? Should the labels be switched?
P4L3-10: I appreciate the discussion of the atmospheric relevance of the samples. However, I feel that the last sentence in this paragraph might better be shifted before "We note…" to introduce the reader to this topic.
P4L12: Change "are" to "were".
P4L21: Change "multiple charged" to "multiply-charged", also in the other instances. Also, "single charged" should become "singly-charged".
P5L12: Remove hyphen in "ice-layer".
P5L18: Insert hyphen in "in line".
Fig. 3: Please state how you derived the error bars.
P6L13: How were the 25 % derived? Which particle size are you referring to?

P6L22-25: Could you state the parameters of the amazonite contact angle distribution? Should amazonite be capitalized on P6L22?

P6L26: Here you could refer to the Fig. showing $n_s(T)$ which I suggested as a replacement for Fig. 5.

P7L7-9: This statement would be more convincing if you provided actual numbers for the remaining multiply-charged particles in the 400 and 500 nm aerosol.

P7L10-11: I advise not to use $T_{50}$ for comparison to other studies. In this regard, my suggestion from above, i.e., showing $n_s(T)$, would be helpful.

P7L23-25: "it remains unknown what particle property other than chemistry and crystallography or morphological features … could be active sites": This conclusion cannot be made at this point in the manuscript since you only discuss these factors in Sec. 5. Please reword. On P7L25, do you mean "as discussed"?

P8L8: "sanidine" should also be followed by a comma.

P8L11: Sometimes you use "(see Figure…)", sometimes only "(Figure…)". Be consistent.

P8L13-14: What is the difference between a defect-free and an ordered crystal? Please clarify.

P8L29: I think, it might be helpful to indicate the perthitic structures in Fig. 6, maybe with the help of overlaid boxes.

P9L29: "Contrary to the Pb content, …": Are you referring to microcline not quite fitting the linear relation in Fig. 7? This should be discussed in the previous paragraph.

P10L4: Move "(increase entropy)" behind "structuring of water".

P10L6-7: The explanations of kosmotropic and chaotropic in brackets should be moved to P10L2, where the terms are first mentioned.

P10L10-11: I suggest to move this statement towards the beginning of Sec. 5.2. Otherwise the reader might wonder for quite some time how valid your conclusions about the bulk chemical composition are for the investigated submicron particles.

P10L28: Insert comma between "cold" and "low".

P11L9: Missing bracket after "sanidine".

P11L25-26: Either change "temperatures" to "a temperature" or "that" to "those".

References:

Augustin-Bauditz, S., Wex, H., Kanter, S., Ebert, M., Niedermeier, D., Stolz, F., Prager, A., and Stratmann, F.: The immersion mode ice nucleation behavior of mineral dusts: A comparison of different pure and surface modified dusts, Geophys. Res. Lett., 41, 7375-7382, doi:10.1002/2014GL061317, 2014.

Harrison, A. D., Whale, T. F., Carpenter, M. A., Holden, M. A., Neve, L., O'Sullivan, D., Vergara Temprado, J., and Murray, B. J.: Not all feldspars are equal: a survey of ice nucleating properties across the feldspar group of minerals, Atmos. Chem. Phys., 16, 10927-10940, doi:10.5194/acp-16-10927-2016, 2016.

Hartmann, S., Wex, H., Clauss, T., Augustin-Bauditz, S., Niedermeier, D., Rösch, M., and Stratmann, F.: Immersion Freezing of Kaolinite: Scaling with Particle Surface Area, J. Atmos. Sci., 73, 263-278, doi:10.1175/JAS-D-15-0057.1, 2016.

Peckhaus, A., Kiselev, A., Hiron, T., Ebert, M., and Leisner, T.: A comparative study of K-rich and Na/Ca-rich feldspar ice-nucleating particles in a nanoliter droplet freezing assay, Atmos. Chem. Phys., 16, 11477-11496, doi:10.5194/acp-16-11477-2016, 2016.

Whale, T. F., Holden, M. A., Kulak, A. N., Kim, Y.-Y., Meldrum, F. C., Christenson, H. K., and Murray, B. J.: The role of phase separation and related topography in the exceptional ice-nucleating ability of alkali feldspars, Phys. Chem. Chem. Phys., 19, 31186-31193, doi:10.1039/C7CP04898J, 2017.

---

## Referee Comment (RC2) · Alexey Kiselev (Referee) · 8 Mar 2019

**Referee report on "Ice nucleation properties of K-feldspar polymorphs and plagioclase feldspars" by André Welti et al.**

The paper "Ice nucleation properties of K-feldspar polymorphs and plagioclase feldspars" by A. Welti, U. Lohmann, and Z. Kanji continues the series of experimental studies aimed at understanding the ice nucleating (IN) properties of the feldspar component of atmospheric mineral dust (MD). In particular, the role of potassium-rich feldspar and its modifications requires a thorough characterization due to its abundance in the atmosphere and the easiness it triggers ice nucleation in supercooled water or supersaturated water vapor.

Apart from reporting the results of freezing experiments, the manuscript suggests that some qualitative correlations between the feldspar genesis and its IN activity could be established. However, as many factors (pressure, temperature of crystallization, cooling rate, presence of volatile fluxes, shearing stress, variations in bulk composition of magmatic fluid, etc.) can influence the structure of the K-feldspar in igneous rocks, the authors do not attempt specifying a mechanism responsible for the strong variation of IN activity of feldspar. Even so, establishing such correlations could provide a useful tool to predict what kind of feldspar might me expected to be a better IN particle. From this point of view, the correlation between median freezing temperature and Rb/Sr ratio appears to be especially promising (more on that below). I therefore support the publication of the manuscript but would like to attract authors' attention to the following issues:

1. The manuscript reports size resolved immersion freezing experiments on 9 new types of feldspar using a single-particle continuous flow diffusion chamber (CFDC), and thus presents a large body of new data that could be used for parameterization of IN activity of feldspar-containing MD aerosols in cloud models. However, the data is not presented in terms of ice nucleating active site (INAS) density, which became standard for reporting the IN properties of aerosol particles over the past years. Although the possibility of such analysis is indicated (end of section 4.1) and obviously has been conducted for amazonite sample (Ickes et al. 2017), it was not extended to other samples and sizes in this manuscript. This is particularly unfortunate, as the experimental conditions (using mobility-selected single feldspar particles as INPs and well-defined thermodynamic conditions in the CFDC) are ideally suited for performing the INAS density-based analysis. This would substantially simplify the comparison of the results with the numerous data of recent IN experiments on feldspar. I strongly recommend including this analysis into the revised version of the manuscript. Additionally, the size distributions of aerosol particles used in the experiments have to be added to the manuscript (as a supplement) or made available for anyone who would be interested to calculate the INAS densities.

2. That the IN properties of all feldspars are size dependent, is convincingly shown by the figure 5. It is also apparent that the T50 values for the smallest particle sizes in each curve lie in the homogeneous freezing range, implying that the T50 values have been calculated for all droplets, frozen both heterogeneously and homogeneously. This doesn't make sense to me. What is the purpose of reporting the median freezing temperature for particles that did not induce freezing of supercooled droplets? Was it not possible to correct the frozen fraction for homogeneously frozen droplets? Again, this is a very strong argument for reporting the INAS densities instead of median freezing temperatures.

2. Another major (critical) comment is that the reported correlations between crystalline structure, trace element composition, and IN activity are not sufficiently supported by statistics. The authors state that IN efficiency is correlated with the degree of ordering of Si and Al atoms in the feldspar framework, implying that there must be a mechanism responsible for this correlation. This mechanism is not discussed in the manuscript, nor is the suggested correlation expressed in any quantitative manner (although such quantity can be derived, see for example, (Smith 1970, Brown and Parsons 1989). I doubt

that such correlation could be of practical use unless a mechanistic explanation is suggested or a quantitative correlation analysis is performed.

3.      With respect to the trace elements composition, the number of investigated samples is definitely too low to draw any particular conclusion. I agree that for the presented choice of samples, the correlation between the IN activity and the ratio of Rb/Sr content is striking. On the other hand, the partitioning of trace elements takes place as feldspar is cooling and forming separate phases, so that Rb replaces K and Sr replaces Ca; thus the Rb/Sr ratio is indicative for the position of feldspar on the ternary phase diagram (An-Ab-Or) and the absolute concentrations of trace elements depends on the cooling/crystallization process and abundance of trace elements in particular magmatic fluid (Parsons et al. 2008). As a consequence, the high Rb content is expected in low microcline and the observed correlation just reflects the correlation between the IN activity and microstructure of feldspar. It is still very interesting that the most IN active sample has also the highest Rb content, an order of magnitude higher than the second-best microcline. I think this fact deserves to be highlighted in the manuscript, even if no mechanistic explanation can be offered at the moment. With the amount of experimental data on IN properties of feldspars that recently became available, it would be relatively easy to improve the statistical analysis and to show if the connection of IN activity and Rb content is not just a coincidence. I would really like to see such analysis in the manuscript or in the follow-up work.

4.      I am also confused by the discussion of Pb content and its correlation with the median freezing temperature. The authors state that "*The best linear correlation between a single compound and the ice nucleation activity based on T50, was found for the Pb content in the feldspar samples (Figure 7).*", but obviously the Pb content in microcline (better IN) is lower than in orthoclase (weaker IN). The quality of the correlation as expressed by Pearson's r-coefficient  is apparently influenced by the low Pb content in the plagioclase and high-temperature disordered alkali feldspar, which has been formed before any trace elements partitioning could take place. I think the plagioclase feldspars should be excluded from the correlation analysis. If you do that, the correlation between T50 and Rb/Sr content ratio would be even better than for Pb and the T50 as a function of Rb/Sr would be also monotonically rising.

        On a side note, if Pb is normalized by Nd content, the correlation between T50 and Pb/Nd ratio becomes very clear, with T50 as a function of Pb / Nd staying constant for plagioclases  and then monotonically rising from the level of adularia and sanidine towards amazonite. This is just to illustrate that such correlations can be constructed very easily and most of them are probably meaningless, unless a larger set of samples is analyzed or an underlying physical and chemical mechanism is suggested. I encourage the authors to do so.

5.      With respect to the potential predictability of IN activity based on the relative concentrations of trace elements, I have to point out that the measurements reported here have been conducted exclusively with the single-crystal samples. In case of real-world atmospheric mineral dust, the majority of feldspar would be coming from phenocrysts (inclusions of feldspar in e.g. granite matrix), where the trace elements content could be very different. Again, the idea of using partitioning of trace elements as a predictor for IN efficacy is very attractive and promising, but must be explored deeper and supported by extensive dataset.

**Specific remarks** (ordered according to page and line number, citations given in italic):

1-14. The sentence "*Ice nucleation is most efficient on the crystallographic ordered, triclinic K-feldspar species microcline, while the intermediate and disordered, monoclinic K-feldspar polymorphs orthoclase and sanidine nucleate ice at lower temperatures.*" strongly implies a causality between the degree of ordering and IN efficacy. Since such causality is not supported by the experimental data of the manuscript, I suggest that the sentence should be reformulated.

1-26 (Introduction). The introduction would greatly benefit from including a discussion of partitioning of the trace elements in feldspars. As mentioned above, the possible correlation between trace elements and IN activity can open a unique opportunity to classify the feldspars on a single-particle basis, for example in a laser ablation mass spectrometer.

2-9. The degree of order (or disorder) of Al3+ and Si4+ is not something that an average atmospheric scientist would be familiar with. This sentence requires explanation.

3-2. (Whale et al. 2017) did not conduct systematic study of crystalline structure in terms of disorder, although they report the fraction of orthoclase in their perthitic samples. A careful analysis of their samples could have reveal a correlation. Please elaborate on that issue.

3-14. (Niedermeier et al. 2015) has reported nucleation ability of size-resolved K-feldspar sample in single-particle immersion mode and according to the manuscript, the microcline data from your study has been previously published in (Ickes et al. 2017). So the statement "for the first time" must be either removed or explained, what exactly has been done in this work for the first time.

3-21 (Section 2). A more detailed description of the sample origin and the XRF analysis would be very helpful here. Was XRF analysis the only basis for identifying the samples or have you performed the powder XRD analysis, too? Could you speculate on how the chemical composition of single aerosol particles would be related to the chemical composition measured for bulk samples?

3-27. The sample that is called "orthoclase" here should be the one closest to the ideal end-member of the alkali feldspar group (Or = KAlSi3O8). On the Figure 1, however, this sample has the same composition as amazonite with almost 20% albite. Where the name "orthoclase" came from?

By the way, it is not correct to label the axes of the ternary phase diagram with fraction of K, Na, or Ca. The ternary diagram gives a sample composition in terms of weight fractions of end-members (orthoclase, albite, and anorthite), see for example (Parsons 2010). The name orthoclase is also misspelled in the legend. Please correct.

4-3. Table 3: What is the purpose of reporting compounds that could not be detected in any of the samples (FeO, NiO, H2O, CO2)? What is the difference between a "0" and N/D?

6-21:22.  This kind of analysis has been published even before (Ickes et al. 2017) and should be mentioned here. See, for example (Wright et al. 2013, Niedermeier et al. 2015, Peckhaus et al. 2016) to name just a few.

7 (Section 4.2) This section should be expanded to include more thorough size dependence analysis, ideally complimented by the INAS density calculations. The T50 should be corrected to account for the homogeneous freezing of droplets containing the smallest particle sizes. Some specific question here: why would one expect linear dependence of T50? The median freezing temperature is a function of cooling rate and residence time, have you taken this into account when comparing T50 from your measurements and data from Atkinson et al., (2013)?

7-9. "*The minimum size triggering immersion freezing is found to be 50 nm microcline particles*". In the Sample preparation section you mention that a "*substantial fraction of larger multiple charged particles… among the 50nm particles can be expected*". How substantial is this fraction? Since the frozen fraction ends in the homogeneous freezing regime at 0.3 (Figure 3), is there a way to decide if the 50 nm microcline particles have been responsible for freezing at all or only the multiply charged larger particles are responsible? On the other hand, the freezing curve for 50 nm amazonite particles reaches value of 0.9 suggesting that all particles have been active. Could that be that even smaller particles would be IN active?

8-25 Twinning is not just symmetrical intergrowth. In alkali feldspars twinning is interrelated with phase exsolution and Na-K exchange between phases or between feldspar and external aqueous fluids (Parsons et al. 2015).

9-4. How thick was the Pt coating applied prior to SEM imaging? For the confinement effects to become important the pores should be less than 10 nm in diameter, could you reach this resolution in the SEM analysis?

9-9. "*The assumption that physical properties (e.g. hardness) are comparable among the tested feldspar species implies that the same degree of artificial surface features are introduced to all samples.*" What is the background for such assumption? Could you support it by literature data? On the page 3 line 32 I read something different: "*The different polymorphs differ in physical properties (cleavage, hardness, specific weight, melting point)…*" Could you clarify this point?

10-5. The discussion of kosmotropic vs. chaotropic cations is not very convincing. Why would the substitution of K+ for Rb+ increase the IN efficiency of K-rich feldspar if K+ is already kosmotropic? Or is there anything known about the degree of "kosmotropicity" for different kosmotropic cations? How many ions of Rb+ would one expect on the surface of an aerosol particle? Why some Na/Ca-rich feldspars exhibit a strong IN activity, having neither K+ nor Rb+ in their structure (like amelia albite in Whale et al., 2017)? This hypothesis should be either discussed in more detail or just omitted from the manuscript.

I am looking forward to the revised version of the manuscript.

References

1.      Brown, W. L. and I. Parsons (1989). "Alkali feldspars: ordering rates, phase transformations and behaviour diagrams for igneous rocks." Mineralogical Magazine **53**(369): 25-42 doi: 10.1180/minmag.1989.053.369.03.

2.      Ickes, L., A. Welti and U. Lohmann (2017). "Classical nucleation theory of immersion freezing: sensitivity of contact angle schemes to thermodynamic and kinetic parameters." Atmos. Chem. Phys. **17**(3): 1713-1739 doi: 10.5194/acp-17-1713-2017.

3.      Niedermeier, D., S. Augustin-Bauditz, S. Hartmann, H. Wex, K. Ignatius and F. Stratmann (2015). "Can we define an asymptotic value for the ice active surface site density for heterogeneous ice nucleation?" Journal of Geophysical Research: Atmospheres: n/a-n/a doi: 10.1002/2014jd022814.

4.      Parsons, I. (2010). Feldspars defined and described: a pair of posters published by the Mineralogical Society. Sources and supporting information. Mineralogical Magazine. **74:** 529.

5.      Parsons, I., J. D. Fitz Gerald and M. R. Lee (2015). "Routine characterization and interpretation of complex alkali feldspar intergrowths." American Mineralogist **100**(5-6): 1277-1303 doi: 10.2138/am-2015-5094.

6.      Parsons, I., C. W. Magee, C. M. Allen, J. M. G. Shelley and M. R. Lee (2008). "Mutual replacement reactions in alkali feldspars II: trace element partitioning and geothermometry." Contributions to Mineralogy and Petrology **157**(5): 663 doi: 10.1007/s00410-008-0358-1.

7.      Peckhaus, A., A. Kiselev, T. Hiron, M. Ebert and T. Leisner (2016). "A comparative study of K-rich and Na/Ca-rich feldspar ice-nucleating particles in a nanoliter droplet freezing assay." Atmos. Chem. Phys. **16**(18): 11477-11496 doi: 10.5194/acp-16-11477-2016.

8.      Smith, J. V. (1970). "Physical properties of order-disorder structures with especial reference to feldspar minerals." Lithos **3**(2): 145-160 doi: https://doi.org/10.1016/0024-4937(70)90070-8.

9.      Whale, T. F., M. A. Holden, A. N. Kulak, Y.-Y. Kim, F. C. Meldrum, H. K. Christenson and B. J. Murray (2017). "The role of phase separation and related topography in the exceptional ice-nucleating ability of alkali feldspars." Physical Chemistry Chemical Physics **19**(46): 31186-31193 doi: 10.1039/C7CP04898J.

10.     Wright, T. P., M. D. Petters, J. D. Hader, T. Morton and A. L. Holder (2013). "Minimal cooling rate dependence of ice nuclei activity in the immersion mode." Journal of Geophysical Research: Atmospheres **118**(18): 10,510-535,543 doi: 10.1002/jgrd.50810.

---

## Author Response (AR1)

**Response to Reviewer 1: Reviewer comments are reproduced in bold font and author comments in regular typeface.**

**Review of "Ice nucleation properties of K-feldspar polymorphs and plagioclase feldspars" by André Welti and co-authors for Atmospheric Chemistry and Physics**

**General comments:**
**Welti et al. present a study about the immersion freezing behavior of a variety of different feldspar samples which builds on recent investigations by Augustin-Bauditz et al. (2014), Peckhaus et al., 2016), Harrison et al. (2016), and Whale et al. (2017). The samples were chosen carefully to provide a variety of crystal structures, chemical compositions, and ordering of the crystal lattice. They include five polymorphs of K-feldspar and four plagioclase feldspars. The immersion freezing experiments were performed with droplets containing single, size-selected particles and care was taken to minimize the amount of multiply-charged particles in the sample aerosol. Furthermore, the study includes X-ray fluorescence measurements giving information about bulk chemical composition and scanning electron microscopy images of particle morphology.**

**What differentiates the present paper from the earlier studies is the discussion of the effect of particle size and the degree of order in the crystal network on the ice nucleation efficiency of the samples. The authors' conclusions concerning these factors are generally comprehensible and well substantiated by the presented results. However, in some cases, which are pointed out in the specific comments, I am missing a more precise explanation. The figures are mostly clear, but I would like to suggest some alternatives for presenting the data (see specific comments). Language-wise, the paper is concisely written but some minor adjustments would increase readability (see technical corrections).**
**Overall, the paper is interesting, understandable, and fits within the scope of Atmospheric Chemistry and Physics. I recommend publication after minor editing.**

We thank the reviewer for the comments and suggestions. We address the specific comments individually below.

**Specific comments:**
**1) Size dependence of ice nucleation behavior**

**The authors often refer to the "pronounced size dependence of ice nucleation activity" as if this is a rarely observed feature. Normally, the ice nucleation behavior scales with the surface area of the immersed particles, meaning that the efficiency increases with increasing particle size. The authors should clarify to what extent the size dependent ice nucleation behavior of their samples deviates from the standard. In my opinion, this is best done by calculating the ice nucleation active surface site density $n_s(T)$ for differently sized particles. In contrast to the chosen $T_{50}$ approach, this method would have the benefit of providing an overview over the whole investigated temperature range. I hence suggest to replace Fig. 5 with a multi-panel figure (like Fig. 3) showing $n_s(T)$ of the different particle sizes for all investigated samples (see, e.g., Fig. 5 in Hartmann et al., 2016).**

We prefer to leave Figure 5 as is, but now add a figure in the appendix for the INAS densities since converting the frozen fraction to INAS densities involves accounting for the surface area, which is easily calculated given the particle size information in the manuscript. Concerning the extent of size dependence, we observe that all samples show a size dependence, mostly even stronger than linearly scaling with surface area with decreasing particle size. One microcline sample shows almost linear scaling of the frozen fraction with surface area. As all samples have this strong size dependence, we indeed consider it "pronounced". To be more specific we changed the sentence to (page 1 line 21-23): "A pronounced size dependence of ice

nucleation activity for the feldspar samples is observed, with the activity of smaller sized particles scaling with surface area or being even higher compared to larger particles. The size dependence varies for different feldspar samples."

**Furthermore, the authors state that "microcline exhibited immersion freezing even for 50 nm particles" whereas for orthoclase "ice nucleation requires active sites present on 400-800 nm sized particles" and relate this observation directly to the effect of these particles on atmospheric ice nucleation. Concerning the potential of these species as atmospheric INP, one must always combine their efficiency and their abundance. Larger orthoclase particles might be needed to trigger ice nucleation as efficiently as smaller microcline particles, but maybe many more orthoclase particles are emitted into the atmosphere?**

This is true, we agree, we have clarified this in the text, that atmospheric abundance of ice nucleation species needs to be considered in order to determine atmospheric relevance (see page 1 line 26-27)

**Besides, the fact that ice nucleation was not observed for 100 or 200 nm orthoclase particles is related to the detection limit of the instrument. If the authors had investigated more droplets, they would eventually have observed ice nucleation triggered by the small orthoclase particles. This should be made clear in the manuscript.**

We disagree with this reasoning, the argument of the detection limit applies to both, the orthoclase and the microcline experiments, the detection limit did not change between these two. i.e. we did not have to increase the amount of microcline in a single droplet, or increase the number of droplets observed in order to observe ice nucleation in the 50 nm microcline sample. The fact that 50 nm microcline (amazonite) demonstrated ice nucleation activity similar to that of 800 nm orthoclase, shows that even for the same detection limit, the 50 nm particles are potent INPs compared to the 100 or 200 nm orthoclase particles. i.e. by changing the number of droplets, or detection limit, this would not change the conclusions that 50 nm microcline particles are much more active than the 100 nm orthoclase particles.

**2) Difference to earlier studies**
**This refers to P3L14-18, where I think the authors should clarify the innovation of their study more. Like this, it sounds as if they might expect an effect of methodology on the results, as the other studies were performed with droplets containing numerous particles each. Please state that by using single particles, you are focusing on a different temperature range than the other studies (except Augustin-Bauditz et al., 2014).**

We have now clarified that we are able to focus on lower temperatures (< 253 K) with the single immersed particles (see page 3 line 30-31 in revised manuscript). However, the observed size dependence also indicates that except for cases where activity scales linearly with surface area (only one Microcline in this study), the particle size distribution used in drop freezing experiments (which make an assumption of linear scaling of activity with surface area) can potentially have an influence on the measured activity.

**3) Multiply-charged particles**
**The authors should be more precise concerning the amount of multiply-charged particles in the cases where the CPMA was not used. This issue could be addressed by including actually measured size distributions in Fig. 2 instead of the schematic ones. Alternatively, the authors could include the following statement on P4L28-29: "The use of the CPMA for the selection of larger particles (400 nm, 800 nm) was not necessary as the fraction of larger particles was reduced to … % by the cyclones and the impactor upstream of the DMA."**

We agree with the reviewer and now include specific numbers for the diameters and fraction of multiply charged particles in section 3.1 of the revised manuscript (see page 5 line 10 -22). We also include a figure in Appendix C to show the contribution of the multiply charged particles to the frozen fractions for the 50 nm sample, where the highest multiple charged particle fraction is calculated.

**4) Figure 4**
**I see more benefit from one figure showing *FF* over *T* for 800 nm particles of all samples. This would be more suited for comparing the ice nucleation signatures of the different feldspars than just showing the range in which freezing occurred. Error bars could be omitted (because they are already shown in Fig. 3) and symbol size reduced for clarity.**

Even without the error bars, such a figure becomes messy and actually rather difficult to read, as there is a huge amount of data overlap. It results in not being able to see the frozen fraction curves for many samples because of the temperature overlap. As such we keep Figure 4 as is. Furthermore, not having error bars if we plot frozen fraction vs. temperature would not allow for a realistic comparison of differences (or similarities) between the samples.

**Technical corrections:**
**P1L9: Replace "Na/Ca-rich feldspar" with plagioclase to be consistent with the title. The composition is explained below anyhow.**
We agree, done (page 1 line 9 revised manuscript)

**P1L11: Replace "are" with "were" in "Samples are selected…".**
Done (page 1 line 11)

**P1L18-20: This sentence would benefit from being split into two.**
We agree, done (page 1, line 18-19).

**P1L24-25: Either omit the "s" at "temperatures" or at "depends".**
We changed "depends" to "depend" (page 1 line 26).

**P1L29: There is also contact freezing in which the contact causes nucleation, not an immersed particle.**
We have now adjusted page 2 line 2 -5 to reflect contact nucleation as a mechanism in this explanation of freezing of supercooled drops.

**P2L12: Within a sentence "e.g." should be preceded by a comma. A comma should also follow in case you are using American English. This also applies to "i.e.". Generally, check your manuscript for consistency with either British or American English. E.g., see "favouring" on P11L10 or "generalise" and "analysed" on P12L8 and L13.**
We inserted the comma before "e.g." (page 2 line 17) but not after as we are using British English as is evident by the words specified by the reviewer.

**P2L13: Less efficient in comparison to which other species?**
Compared to other mineral species such as muscovite and kaolinite – we have now clarified this aspect on page 2 line 18 of the revised manuscript.

**P2L14: Change "for example" to "e.g.".**
Done, now page 2 line 19

**P2L22: Capital "X" in "x-ray".**
Done, now page 2 line 27

**P3L21: Mention that XRF is a bulk, not a single particle technique.**
We now mention this on page 4 line 3, in addition we note that we explicitly state that XRF is a bulk composition measurement when discussing the data in section 5.2, already in the original manuscript.

**P3L28: Omit comma following "polymorphism" and add "s" to "occur".**
Done, now page 4 line 10

**P3L30-31: Add comma behind "sanidine". Be consistent using either "temperature" or "temperatures".**
Comma added, and "temperatures" corrected to "temperature", page 4 line 12 in revised manuscript.

**P4L1-2: I suggest to remove the brackets and structure the sentence as follows: "sanidine in volcanic and very high-temperature metamorphic rocks, orthoclase in … rocks and microcline in … rocks."**
Done! Page 4 line 14-16 revised manuscript.

**P4L2: "feldspar": This should be plural.**
Changed to plural and added comma before feldspars (page 4 line 16)

**P4L3: Why is Table 3 referred to before Table 2 is mentioned? Should the labels be switched?**
Thank you for pointing this out. We now refer to section 5.2 (page 4 line 17) instead of Table 2.

**P4L3-10: I appreciate the discussion of the atmospheric relevance of the samples. However, I feel that the last sentence in this paragraph might better be shifted before "We note…" to introduce the reader to this topic.**
This has now been done. The last sentence "Samples used in this study.." has now been moved further up in the paragraph (Page 4 line 17-19 in revised manuscript). "We note" has now been changed to "As such.." (page 4 line 19)

**P4L12: Change "are" to "were".**
Done, page 4 line 26

**P4L21: Change "multiple charged" to "multiply-charged", also in the other instances. Also, "single charged" should become "singly-charged".**
This is a matter of preference, as such there is no rule saying multiple should be multiply etc. As such we retain the current structure. However, we note the lack of plurals on page 5 lines 4-5, and have corrected those.

**P5L12: Remove hyphen in "ice-layer".**
Done (now page 5 line 31)

**P5L18: Insert hyphen in "in line".**
Done (now page 6 line 4)

**Fig. 3: Please state how you derived the error bars.**
We have added a description of the error bars in the caption of Figure 3 and refer the interested reader to the work of Lüönd et al. (2010).

**P6L13: How were the 25 % derived? Which particle size are you referring to?**
Thanks for catching that, we were referring to the 800 nm particle curves in Figure 3. We have now corrected this (page 6 line 29).

**P6L22-25: Could you state the parameters of the amazonite contact angle distribution? Should amazonite be capitalized on P6L22?**
The parameters are the mean contact angle, and the variance of the distribution. However, the contact angle distribution referred to in page 7 line 11 is from a different paper (Ickes et al., 2017) and bears no inclusion here for just a single sample and is also not the focus or objective of the paper. We corrected the capitalization of amazonite on page 7 line 8 (revised manuscript) the reviewer is correct, this does not require capitalization. Thanks!

**P6L26: Here you could refer to the Fig. showing $n_s(T)$ which I suggested as a replacement for Fig. 5.**
We now refer to the INAS figure A1 in the appendix A (page 6, line 12-13).

**P7L7-9: This statement would be more convincing if you provided actual numbers for the remaining multiply-charged particles in the 400 and 800 nm aerosol.**
We have now provided the fraction of multiply charged aerosol for all the sizes in section 3.1 (see page 5 line 10-21). However, we would like to re-iterate that the fraction of multiply charged particles have no bearing on this statement, as the convergence is occurring at the higher surface areas (i.e. larger particles sizes) where we have the fewest percentage of multiple charged particles (see section 3.1 of revised manuscript).

**P7L10-11: I advise not to use $T_{50}$ for comparison to other studies. In this regard, my suggestion from above, i.e., showing $n_s(T)$, would be helpful.**
We have included $n_s(T)$ (we use the term INAS density) in the appendix following the Reviewer recommendations. It is not clear from the Reviewer's comment why in addition a comparison using $T_{50}$ cannot be presented.

**P7L23-25: "it remains unknown what particle property other than chemistry and crystallography or morphological features … could be active sites": This conclusion cannot be made at this point in the manuscript since you only discuss these factors in Sec. 5. Please reword. On P7L25, do you mean "as discussed"?**
Agreed, we have rephrased the statement, which is meant to introduce and explain the structuring of the discussion and not to draw any conclusions (see page 8 line 10-12).

**P8L8: "sanidine" should also be followed by a comma.**
Done (page 8 line 27)

**P8L11: Sometimes you use "(see Figure…)", sometimes only "(Figure…)". Be consistent.**
This is not accidental; there is no need for consistency here, because the two indicate different things. When we say "see Figure xx" we are suggesting that a reader should look at the Figure while reading that sentence, and when we just say (Figure xx) we are informing the reader where the relevant information is available.

**P8L13-14: What is the difference between a defect-free and an ordered crystal? Please clarify.**
The degree of order and disorder in feldspars is determine by the distribution of silicon and aluminum distribution within the tetrahedrons, in sanidine the distribution is random, where as in an ordered crystal (microcline), the distribution of these atoms are regular. Defects can occur in both, ordered or disordered crystals, since a defect would imply a point defect or a line defect. In the former, this can be because of a

vacancy at a point where there should be an atom, or the presence of an atom at a location where there should be empty (interstitial space). Line defects can occur if atoms are misaligned

**P8L29: I think, it might be helpful to indicate the perthitic structures in Fig. 6, maybe with the help of overlaid boxes.**
We think this could reduce the aesthetic of the images. But we add a description to the caption in Figure 6 and refer to it at the said location now page 9 line 20-23

**P9L29: "Contrary to the Pb content, …": Are you referring to microcline not quite fitting the linear relation in Fig. 7? This should be discussed in the previous paragraph.**
Done (see page 10 line 15-16 and line 23-24).

**P10L4: Move "(increase entropy)" behind "structuring of water".**
Done (page 11 line 8)

**P10L6-7: The explanations of kosmotropic and chaotropic in brackets should be moved to P10L2, where the terms are first mentioned.**
Done (now page 11 line 6)

**P10L10-11: I suggest to move this statement towards the beginning of Sec. 5.2. Otherwise the reader might wonder for quite some time how valid your conclusions about the bulk chemical composition are for the investigated submicron particles.**
Done. We moved this sentence to page 10 line 9-10 at the beginning of section 5.2. In addition, we also refer to the INAS densities in appendix A to demonstrate the similarity in composition with particle size.

**P10L28: Insert comma between "cold" and "low".**
Done, (page 12, line 1).

**P11L9: Missing bracket after "sanidine".**
Done – thanks (page 12 line 14).

**P11L25-26: Either change "temperatures" to "a temperature" or "that" to "those".**
We restructured this sentence to "above homogeneous freezing temperatures" so that the plural use is more obvious (page 12 line 32)

**References:**
**Augustin-Bauditz, S., Wex, H., Kanter, S., Ebert, M., Niedermeier, D., Stolz, F., Prager, A., and Stratmann, F.: The immersion mode ice nucleation behavior of mineral dusts: A comparison of different pure and surface modified dusts, Geophys. Res. Lett., 41, 7375-7382, doi:10.1002/2014GL061317, 2014.**

**Harrison, A. D., Whale, T. F., Carpenter, M. A., Holden, M. A., Neve, L., O'Sullivan, D., Vergara Temprado, J., and Murray, B. J.: Not all feldspars are equal: a survey of ice nucleating properties across the feldspar group of minerals, Atmos. Chem. Phys., 16, 10927-10940, doi:10.5194/acp-16-10927-2016, 2016.**

**Hartmann, S., Wex, H., Clauss, T., Augustin-Bauditz, S., Niedermeier, D., Rösch, M., and Stratmann, F.: Immersion Freezing of Kaolinite: Scaling with Particle Surface Area, J. Atmos. Sci., 73, 263-278, doi:10.1175/JAS-D-15-0057.1, 2016.**

**Peckhaus, A., Kiselev, A., Hiron, T., Ebert, M., and Leisner, T.: A comparative study of K-rich and Na/Ca-rich feldspar ice-nucleating particles in a nanoliter droplet freezing assay, Atmos. Chem. Phys., 16, 11477-11496, doi:10.5194/acp-16-11477-2016, 2016.**

**Whale, T. F., Holden, M. A., Kulak, A. N., Kim, Y.-Y., Meldrum, F. C., Christenson, H. K., and Murray, B. J.: The role of phase separation and related topography in the exceptional ice-nucleating ability of alkali feldspars, Phys. Chem. Chem. Phys., 19, 31186-31193, doi:10.1039/C7CP04898J, 2017.**

References

Lüönd, F., Stetzer, O., Welti, A., and Lohmann, U.: Experimental study on the ice nucleation ability of size-selected kaolinite particles in the immersion mode, J. Geophys. Res.-Atmos., 115, DOI:10.1029/2009jd012959, 2010.

**Reviewer comments have been reproduced in bold and author responses in regular typeface.**

**Referee report on "Ice nucleation properties of K-feldspar polymorphs and plagioclase feldspars" by André Welti et al. The paper "Ice nucleation properties of K-feldspar polymorphs and plagioclase feldspars" by A. Welti, U. Lohmann, and Z. Kanji continues the series of experimental studies aimed at understanding the ice nucleating (IN) properties of the feldspar component of atmospheric mineral dust (MD). In particular, the role of potassium-rich feldspar and its modifications requires a thorough characterization due to its abundance in the atmosphere and the easiness it triggers ice nucleation in supercooled water or supersaturated water vapor.**

**Apart from reporting the results of freezing experiments, the manuscript suggests that some qualitative correlations between the feldspar genesis and its IN activity could be established. However, as many factors (pressure, temperature of crystallization, cooling rate, presence of volatile fluxes, shearing stress, variations in bulk composition of magmatic fluid, etc.) can influence the structure of the K-feldspar in igneous rocks, the authors do not attempt specifying a mechanism responsible for the strong variation of IN activity of feldspar. Even so, establishing such correlations could provide a useful tool to predict what kind of feldspar might me expected to be a better IN particle. From this point of view, the correlation between median freezing temperature and Rb/Sr ratio appears to be especially promising (more on that below). I therefore support the publication of the manuscript but would like to attract authors' attention to the following issues:**

**1. The manuscript reports size resolved immersion freezing experiments on 9 new types of feldspar using a single-particle continuous flow diffusion chamber (CFDC), and thus presents a large body of new data that could be used for parameterization of IN activity of feldspar-containing MD aerosols in cloud models. However, the data is not presented in terms of ice nucleating active site (INAS) density, which became standard for reporting the IN properties of aerosol particles over the past years. Although the possibility of such analysis is indicated (end of section 4.1) and obviously has been conducted for amazonite sample (Ickes et al. 2017), it was not extended to other samples and sizes in this manuscript. This is particularly unfortunate, as the experimental conditions (using mobility-selected single feldspar particles as INPs and well-defined thermodynamic conditions in the CFDC) are ideally suited for performing the INAS density-based analysis. This would substantially simplify the comparison of the results with the numerous data of recent IN experiments on feldspar. I strongly recommend including this analysis into the revised version of the manuscript. Additionally, the size distributions of aerosol particles used in the experiments have to be added to the manuscript (as a supplement) or made available for anyone who would be interested to calculate the INAS densities.**

We added the figure below as Appendix A in the revised manuscript showing INAS densities. Compared to the experimental data shown in Fig. 3 of the manuscript, INAS density ($n_s$) scales the frozen fraction (FF) by the particle surface area according to:

$$n_s = \frac{ln(1 - FF)}{A}$$

Where $A$ denotes the geometric surface area, calculated as $A = 4\pi \left(\frac{d}{2}\right)^2$, with d being the selected mobility diameter. Because measurements are conducted with quasi monodisperse particles, INAS densities can easily be calculated without the need for individual size distributions.

[Figure]

Figure A1 (in revised manuscript): Ice nucleation sites per surface area (INAS) of 9 feldspar samples as a function of temperature and size corresponding to observed frozen fractions as shown in Figure 3. The upper (80-100%) and lower (0-20%) frozen fractions have been omitted to exclude saturation errors from the detection.

**2. That the IN properties of all feldspars are size dependent, is convincingly shown by the figure 5. It is also apparent that the T50 values for the smallest particle sizes in each curve lie in the homogeneous freezing range, implying that the T50 values have been calculated for all droplets, frozen both heterogeneously and homogeneously. This doesn't make sense to me. What is the purpose of reporting the median freezing temperature for particles that did not induce freezing of supercooled droplets? Was it not possible to correct the frozen fraction for homogeneously frozen droplets? Again, this is a very strong argument for reporting the INAS densities instead of median freezing temperatures.**

In cases when homogeneous and heterogeneous nucleation contribute to ice formation, it is beyond the experimental capacity to accurately determine the corresponding fractions. When the measured $T_{50}$ lies within homogeneous freezing temperatures, all that can be deduced with confidence is that the corresponding sample is not an efficient ice nucleator in the immersion mode at temperatures above homogenous freezing temperature. In response to the Reviewer request, we instead removed the $T_{50}$ data from Figure 5 if a frozen fraction of 50% was not reached above the homogeneous freezing temperature. As requested by the Reviewer, INAS densities have now been included in the appendix (see comment above). It can be seen from these plots that normalizing frozen fractions by the particle surface area in the homogeneous nucleation regime, does not produce overlapping (except for amazonite), but rather

parallel displaced INAS densities. This is an artifact of the *INAS* scaling, treating homogeneous freezing that is not dependent on the size of immersed particles, the same as heterogeneous freezing.

**3. Another major (critical) comment is that the reported correlations between crystalline structure, trace element composition, and IN activity are not sufficiently supported by statistics. The authors state that IN efficiency is correlated with the degree of ordering of Si and Al atoms in the feldspar framework, implying that there must be a mechanism responsible for this correlation. This mechanism is not discussed in the manuscript, nor is the suggested correlation expressed in any quantitative manner (although such quantity can be derived, see for example, (Smith 1970, Brown and Parsons 1989). I doubt that such correlation could be of practical use unless a mechanistic explanation is suggested or a quantitative correlation analysis is performed.**

In the initial manuscript, we already explain that we cannot give a reason for the correlation of crystal structures and the ice nucleation activity of feldspars, since the crystal structures do not match that of ice. We agree the explanations given in the manuscript are qualitative (not supported by statistics) as pointed out by Reviewer 2. However, in the reference given by the reviewer (Smith, 1970), the author also confirms that in theory a quantitative description of order-disorder of crystal structure is possible but this has not been done quantitatively as it poses additional complexity and challenges. As such even in Brown and Parsons (1989), the second reference suggested by the reviewer, only a qualitative description is suggested. Quantifying the degree of order in feldspar samples would likely be a full study of its own and much beyond the scope of this work.

The intention of discussing crystalline structure and trace elemental composition is to highlight the possibility that several feldspar properties are connected (e.g. by the conditions of rock formation), and more detailed investigations are needed to disentangle their importance for the ice nucleation activity and the mechanism. Because the lack of a mechanistic understanding we prefer not to provide statistical correlations that could be misunderstood as parametrizations and hope this study to serve as a stepping stone for future studies aiming to establish a quantitative understanding.

Nevertheless, we suggest that it is possible to identify properties which can be used as qualitative tracers (not causes) for the ice nucleation activity. Focus is therefore placed on the most practical tracer, which we think is the trace elemental composition, for which qualitative agreement is shown in Figures 7 and 8.

**3. With respect to the trace elements composition, the number of investigated samples is definitely too low to draw any particular conclusion. I agree that for the presented choice of samples, the correlation between the IN activity and the ratio of Rb/Sr content is striking. On the other hand, the partitioning of trace elements takes place as feldspar is cooling and forming separate phases, so that Rb replaces K and Sr replaces Ca; thus the Rb/Sr ratio is indicative for the position of feldspar on the ternary phase diagram (An-Ab-Or) and the absolute concentrations of trace elements depends on the cooling/crystallization process and abundance of trace elements in particular magmatic fluid (Parsons et al. 2008). As a consequence, the high Rb content is expected in low microcline and the observed correlation just reflects the correlation between the IN activity and microstructure of feldspar. It is still very interesting that the most IN active sample has also the highest Rb content, an order of magnitude higher than the second-best microcline. I think this fact deserves to be highlighted in the manuscript, even if no mechanistic explanation can be offered at the moment. With the amount of experimental data on IN properties of feldspars that recently became available, it would be relatively easy to improve the statistical analysis and to show if the connection of IN activity and Rb content is not just a coincidence. I would really like to see such analysis in the manuscript or in the follow-up work.**

We fully agree with the reviewer that we should highlight better that a replacement of K by Rb results in a lower microcline polymorph, yet a higher Rb/Sr ratio, thus a higher ice nucleation activity suggesting that observed correlation reflects a potential relationship between ice nucleation activity and feldspar microstructure. We also agree, with more samples than those used here, a systematic analysis of available data in light of the findings in this manuscript could be a promising follow-up study. Following the Reviewer recommendation, we added:

Page 10 line 31-page 11 line 3:"The Rb/Sr ratio is indicative for the position of feldspar on the ternary phase diagram (Figure 1) and the absolute concentrations of trace elements depends on the cooling/crystallization process and abundance of trace elements in particular magmatic fluid (Parsons et al., 2009). The replacement of K by Rb results in a lower microcline polymorph, and a higher Rb/Sr ratio, correlating to higher ice nucleation activity. This suggests that the observed correlation reflects a possible relationship between ice nucleation activity and feldspar microstructure."

Page 12, line 20-23: "More ice nucleation active samples show higher Rb/Sr ratios, e.g., the most active microcline sample has an order of magnitude higher Rb/Sr ratio than the second-best microcline. Therefore, Rb/Sr ratios could serve as tracer for highly ice nucleation active feldspar particles or even help to differentiate sources of feldspars acting as INP at specific temperatures."

**4. I am also confused by the discussion of Pb content and its correlation with the median freezing temperature. The authors state that "The best linear correlation between a single compound and the ice nucleation activity based on T50, was found for the Pb content in the feldspar samples (Figure 7).", but obviously the Pb content in microcline (better IN) is lower than in orthoclase (weaker IN). The quality of the correlation as expressed by Pearson's r-coefficient is apparently influenced by the low Pb content in the plagioclase and high-temperature disordered alkali feldspar, which has been formed before any trace elements partitioning could take place. I think the plagioclase feldspars should be excluded from the correlation analysis. If you do that, the correlation between T50 and Rb/Sr content ratio would be even better than for Pb and the T50 as a function of Rb/Sr would be also monotonically rising.**

**On a side note, if Pb is normalized by Nd content, the correlation between T50 and Pb/Nd ratio becomes very clear, with T50 as a function of Pb/Nd staying constant for plagioclases and then monotonically rising from the level of adularia and sanidine towards amazonite. This is just to illustrate that such correlations can be constructed very easily and most of them are probably meaningless, unless a larger set of samples is analyzed or an underlying physical and chemical mechanism is suggested. I encourage the authors to do so.**

The decision to compare $T_{50}$ to Pb, Rb and Sr is based on the random forest analysis which identified these three as the most promising predictors. The $r$-coefficients in the text, give the correlation between these elements and not to the $T_{50}$ as the Reviewer states, as such we will clarify the statement so that a potential reader does not mis-understand the correlation. Furthermore, we share the view of the Reviewer that the correlation of Pb content and $T_{50}$ is biased by the low content in the plagioclase feldspars, the limited number of samples and potential sub sequential uptake. It is because of Pb is suggested by the random forest analysis that we discuss it and then argue to discard it as a predictor. We thank the Reviewer for pointing out the possibility to create a monotonic correlation between $T_{50}$ and Pb/Nd. However, in contrast to Rb/Sr, we did not find any literature suggesting an interpretation of this ratio.

To clarify we now write:

Page 10, line 24-25 "Based on these arguments and that the correlation of Pb content to $T_{50}$ is not monotonic (e.g., lower Pb content in microcline than orthoclase, Figure 7) we focus on other predictors suggested by the Random Forest analysis.

Page 11 line 3-5: "We note that a monotonic correlation between $T_{50}$ and a Pb/Nd ratio can also be constructed, but no interpretation of the implication of such a ratio could be found in the literature and neither was Nd suggested as a predictor by the Random Forest analysis."

**5. With respect to the potential predictability of IN activity based on the relative concentrations of trace elements, I have to point out that the measurements reported here have been conducted exclusively with the single-crystal samples. In case of real-world atmospheric mineral dust, the majority of feldspar would be coming from phenocrysts (inclusions of feldspar in e.g. granite matrix), where the trace elements content could be very different. Again, the idea of using partitioning of trace elements as a predictor for IN efficacy is very attractive and promising, but must be explored deeper and supported by extensive dataset.**

We already discuss this limitation of the study in regard to atmospheric aging in the initial manuscript page 10 lines 20-24 (page 11 lines 26-30 in revised manuscript) but we clarify this further prompted by the Reviewer comment and added the following to the revised manuscript:

Page 11, line 31-32: "Additionally, a majority of airborne feldspar particles could be coming from phenocrysts (inclusions of feldspar in e.g. granite matrix), where the trace element content could be different."

**Specific remarks (ordered according to page and line number, citations given in italic):**
**1-14. The sentence "Ice nucleation is most efficient on the crystallographic ordered, triclinic K-feldspar species microcline, while the intermediate and disordered, monoclinic K-feldspar polymorphs orthoclase and sanidine nucleate ice at lower temperatures." strongly implies a causality between the degree of ordering and IN efficacy. Since such causality is not supported by the experimental data of the manuscript, I suggest that the sentence should be reformulated.**

We agree with the Reviewer and in order to limit the statement we now write (Page 1 line 14):
"Amongst the investigated samples, ice nucleation is most efficient on the crystallographic ordered, triclinic K-feldspar species microcline, while the intermediate and disordered, monoclinic K-feldspar polymorphs orthoclase and sanidine nucleate ice at lower temperatures."

**1-26 (Introduction). The introduction would greatly benefit from including a discussion of partitioning of the trace elements in feldspars. As mentioned above, the possible correlation between trace elements and IN activity can open a unique opportunity to classify the feldspars on a single-particle basis, for example in a laser ablation mass spectrometer.**

We think it would be premature to discuss the partitioning of specific trace elements in the introduction, as the correlation of the trace elements to the ice nucleation activity is shown as a result of the current study. However, prompted by the Reviewer's suggestion, we do include a paragraph on trace elements and their importance to feldspar microtexture and characteristics, in the introduction (see page 3 lines 19-27). Additionally, we have added discussions on partitioning of the trace elements in the following places: Page 10 lines 32 to page 11 line 5 and Page 12 lines 20=23.

**2-9. The degree of order (or disorder) of Al3+ and Si4+ is not something that an average atmospheric scientist would be familiar with. This sentence requires explanation.**

We elaborate further on this topic on page 2 line 14-15:
"The four units are made up of one $Al^{3+}$ and three $Si^{4+}$ ions. In a disordered state $Al^{3+}$ can be found in any one of the four tetrahedral sites while in an ordered state $Al^{3+}$ occupies the same site throughout the crystal (Nesse, 2016)."

**3-2. (Whale et al. 2017) did not conduct systematic study of crystalline structure in terms of disorder, although they report the fraction of orthoclase in their perthitic samples. A careful analysis of their samples could have reveal a correlation. Please elaborate on that issue.**

Whale et al. 2017 do discuss order-disorder in their article. In the supplementary material they give a comprehensive summary on the order-disorder of K-feldspar and they have measured it by Raman spectroscopy for some of their samples. It is mentioned that it has been under discussion in relation to ice nucleation ability in other studies. By pointing to examples where disordered feldspar samples showed high ice nucleation activity when measured with their setup they conclude that it is not a necessary feature. However, different from the current study, their experimental setup is sensitive to much rarer particle features (ice active at much higher temperatures) than a single particle investigation, thus the conclusions might not be transferrable.

**3-14. (Niedermeier et al. 2015) has reported nucleation ability of size-resolved K-feldspar sample in single-particle immersion mode and according to the manuscript, the microcline data from your study has been previously published in (Ickes et al. 2017). So the statement "for the first time" must be either removed or explained, what exactly has been done in this work for the first time.**

We removed the statement (Page 3 line 29).

**3-21 (Section 2). A more detailed description of the sample origin and the XRF analysis would be very helpful here. Was XRF analysis the only basis for identifying the samples or have you performed the powder XRD analysis, too? Could you speculate on how the chemical composition of single aerosol particles would be related to the chemical composition measured for bulk samples?**

Composition of the samples were additionally determined by Rietveld refinement of powder XRD pattern. Note that this method is subjective to the fitting procedure to some degree. Results are given below and have been added as Appendix B in in the manuscript.

| Sample | orthoclase | adularia | sanidine | microcline | plagioclase | quartz | Others |
|---|---|---|---|---|---|---|---|
| Orthoclase | 70 | | | | 15 | 5 | 10 |
| Adularia | | 100 | | | | | |
| Sanidine | | | 100 | | | | |
| Microcline | | | | 90 | 10 | | |
| Microcline (Amazonite) | | | | 77 | 22 | 1 | |
| Labradorite (AU) | | | | | 59 | 12 | 29 |
| Pericline | | | | | 84 | 16 | |
| Labradorite CH2 | | | | | 100 | | |
| Labradorite CH1 | | | | | 100 | | |

Samples were provided from the geological collection of ETH Zürich except Labradorite (AU) which is a commercial sample sold in powder form as supplement for ceramics and Amazonite which was provided by a private collector. Because for all except Labradorite (AU), fine particles are obtained by grinding single stones, we expect the bulk chemical analysis to be a good approximation for the majority of particles. However, this was not investigated experimentally and certain compounds may be more abundant at a certain particle size.

**3-27. The sample that is called "orthoclase" here should be the one closest to the ideal end-member of the alkali feldspar group (Or = KAlSi3O8). On the Figure 1, however, this sample has the same composition as amazonite with almost 20% albite. Where the name "orthoclase" came from? By the way, it is not correct to label the axes of the ternary phase diagram with fraction of K, Na, or Ca. The ternary diagram gives a sample composition in terms of weight fractions of end-members (orthoclase, albite, and anorthite), see for example (Parsons 2010). The name orthoclase is also misspelled in the legend. Please correct.**

The sample was identified as orthoclase by experts and subsequent analysis (see table B1 in revised manuscript) confirmed orthoclase as the primary component. Lacking a more correct alternative, we kept the name. We have in addition added the end member names to the ternary phase diagram. We have kept the axis with fraction of K, Ca and Na since we used this analysis from the XRF to place the samples onto the ternary phase diagram as mentioned in the caption of Figure 1 already in the initial manuscript. We corrected the spelling and Figure 1 accordingly.

**4-3. Table 3: What is the purpose of reporting compounds that could not be detected in any of the samples (FeO, NiO, H2O, CO2)? What is the difference between a "0" and N/D?**

The respective compounds have been removed. "0" indicates a reliable measurement of absence of a compound, while N/D stands for not detected.

**6-21:22. This kind of analysis has been published even before (Ickes et al. 2017) and should be mentioned here. See, for example (Wright et al. 2013, Niedermeier et al. 2015, Peckhaus et al. 2016) to name just a few.**

We added to Page 7 line 9-10: "Parametrizations for other feldspar samples can be found in e.g., Niedermeier et al., 2015; Peckhaus et al., 2016"
Wright et al., 2013 did not report a feldspar parametrization.

**7 (Section 4.2) This section should be expanded to include more thorough size dependence analysis, ideally complimented by the INAS density calculations. The T50 should be corrected to account for the homogeneous freezing of droplets containing the smallest particle sizes. Some specific question here: why would one expect linear dependence of T50? The median freezing temperature is a function of cooling rate and residence time, have you taken this into account when comparing T50 from your measurements and data from Atkinson et al., (2013)? 7-9. "The minimum size triggering immersion freezing is found to be 50 nm microcline particles". In the Sample preparation section you mention that a "substantial fraction of larger multiple charged particles… among the 50nm particles can be expected". How substantial is this fraction? Since the frozen fraction ends in the homogeneous freezing regime at 0.3 (Figure 3), is there a way to decide if the 50 nm microcline particles have been responsible for freezing at all or only the multiply charged larger particles are responsible? On the other hand, the**

**freezing curve for 50 nm amazonite particles reaches value of 0.9 suggesting that all particles have been active. Could that be that even smaller particles would be IN active?**

A figure showing the calculated *INAS* density has been included in the appendix. Data points where homogenous freezing strongly contributes to the 50% frozen fraction have been deleted from Figure 5.

A linear dependence of $T_{50}$ in a lin-log plot is expected for CNT based parametrizations using single contact angle or a log-normal contact angle distribution (Welti et al., 2012, Fig. 10).

We did not apply a correction for time dependence when comparing to Atkinson et al., 2013. This is not necessary as both the residence time in the current experiment (12s) and the cooling rate in the Atkinson et al., 2013 experiment (1K/min) maintain the sample at the reported temperature for a comparable duration, i.e. experimental uncertainties outweigh time dependence.

The multiple charged fraction of selected 50nm, amounts to 30% doubly charged, 73nm and 7% triple charged, 91nm particles, while 63% of particles are 50nm. . Based on the size dependence of immersion freezing and the abundance of multiple charged particles, the contribution of the 3 particle sizes at different frozen fractions can be calculated. The result is shown in the Figure below (and now shown and explained in the revised manuscript Appendix C).

[Figure]

We now clarify this aspect on page 5 line 21 and page 8 line 4-5.

50nm particles contribute more than 40% of ice active particles at any frozen fraction. Based on the current measurements it seems plausible that for some microcline species, particles smaller than 50nm are ice active. But it is difficult to produce such small particles.

**8-25 Twinning is not just symmetrical intergrowth. In alkali feldspars twinning is interrelated with phase exsolution and Na-K exchange between phases or between feldspar and external aqueous fluids (Parsons et al. 2015).**

The section was changed accordingly and the reference was added (see page 9 line 14-16 in the revised manuscript).

**9-4. How thick was the Pt coating applied prior to SEM imaging? For the confinement effects to become**

important the pores should be less than 10 nm in diameter, could you reach this resolution in the SEM analysis?

Sputtered films for SEM typically have a thickness range of 2–20 nm. The resolution of the SEM is 1 -1.7 nm. As no pores are detected in the SEM images, and smaller pores that could have been missed in the SEM analysis and are small enough to supress ice formation, we conclude that using the geometric surface is a better measure for the particle surface area than the BET-surface.

**9-9. "The assumption that physical properties (e.g. hardness) are comparable among the tested feldspar species implies that the same degree of artificial surface features are introduced to all samples." What is the background for such assumption? Could you support it by literature data? On the page 3 line 32 I read something different: "The different polymorphs differ in physical properties (cleavage, hardness, specific weight, melting point)…" Could you clarify this point?**

Feldspar do have very similar properties. As an example, their hardness is typically between 6 - 6.5 on Mohrs scale. https://en.wikipedia.org/wiki/Feldspar

We changed the sentence p.3, line 32 (now page 4 line 14)  to "The different polymorphs differ in some physical properties (e.g. melting point) and are found in different rocks:…"

**10-5. The discussion of kosmotropic vs. chaotropic cations is not very convincing. Why would the substitution of K+ for Rb+ increase the IN efficiency of K-rich feldspar if K+ is already kosmotropic? Or is there anything known about the degree of "kosmotropicity" for different kosmotropic cations? How many ions of Rb+ would one expect on the surface of an aerosol particle? Why some Na/Ca-rich feldspars exhibit a strong IN activity, having neither K+ nor Rb+ in their structure (like amelia albite in Whale et al., 2017)? This hypothesis should be either discussed in more detail or just omitted from the manuscript.**

We clarify this discussion as suggested by the reviewer, but keep it in the manuscript as a possible second explanation, for the observed correlation between Rb/Sr and the ice nucleation activity of the feldspars (page 11 lines 3-13). To relativize the importance we added (page 11 line 13-15): "It would be necessary to quantify the concentration of cations required to influence water ordering to determine how influential $Rb^+$ and $Sr^{2+}$ are for ice nucleation given their trace elemental composition.

**I am looking forward to the revised version of the manuscript.**

[revised manuscript text omitted]